

# The importance of city trees for reducing net rainfall: comparing measurements and simulations

Vincent Smets[a*] ,Charlotte Wirion[b*], Willy Bauwens[b] , Martin Hermy[a] , Ben Somers[a], Boud Verbeiren[b]

**[a]** *Division Forest, Nature and Landscape, Department of Earth and Environmental Sciences, KU Leuven, Celestijnenlaan 200E, 3001 Heverlee, Belgium.*

**[b]** *Department of Hydrology and Hydraulic Engineering, Vrije Universiteit Brussel, Pleinlaan 2, BE-1050 Brussels, Belgium.*

*\* Correspondance first author emails:* vincent.smets@kuleuven.be and charlotte.wirion@vub.be

**Abstract**. An in-situ tree interception experiment was conducted to determine the hydrological impact of a solitary standing
Norway maple and small leaved lime in an urban environment. During the two-year experiment, rainfall data was collected and divided into interception, throughfall and stemflow. With approximately 38 % of the gross precipitation intercepted by both trees, the interception storage was higher than for similar studies done in Mediterranean regions. A regression analyses for the Norway maple found rainfall duration, rainfall amount and the tree's leaf area index to be the most important variables influencing interception storage. The regression analysis and the tree interception models by Gash and Rutter, as well as an
adapted version of the Water and Energy Transfer between Soil, Plants and Atmosphere model (WetSpa), were tested for their accuracy in modelling the measured interception storage. The models in general overestimated interception storage for small interception events (< interception storage) and underestimated interception storage for bigger interception events (> interception storage). The regression analysis wasn't stable throughout seasons, event sizes and trees, making it unsuitable for generic use. The method of Gash slightly overperformed WetSpa and Rutter for all events throughout seasons and trees.
However, WetSpa showed a better performance for rainfall events > 10 mm. A scenario analysis, featuring the construction of student houses on a university campus, demonstrated the potential of urban trees to retain rainfall water. Even though trees alone could not restore the natural hydrological balance, they could partly mitigate the increased runoff volume and peak discharge caused by sealing of the natural surface through decreasing the net rainfall that reaches the ground. This study highlights the role of solitary trees in an urban environment where natural hydrological processes are severely altered.

## 1.Introduction
### 1.1 The context

Currently 54 % of the population is living in an urban environment, with an expected increase to 66 % by 2050 (United Nations, Departement of Economic and Social Affairs, 2014). The migration of the growing population towards cities gives rise to a
whole new set of challenges. An urban environment exhibits built-up areas that significantly alter the natural processes (Grimm et al., 2008). This leads to problems such as the urban heat island effect (UHI) (Lauwaet et al., 2015) and the increased density of particle matter (Zhang et al., 2015). Another prominent problem modern cities face is the increase in runoff during and after rain events (Paul & Meyer, 2001). Due to urban expansion and the use of impervious materials such as concrete and asphalt, the hydrological cycle is altered and natural processes such as infiltration and interception are impeded. This results in an
increased runoff that causes significant economic losses, especially during heavy rainfall events. Flood frequency in Western Europe has increased six fold since 1970 (Barredo, 2006) and affected over 3 million people between 1998 and 2009, resulting in an economic loss of 60 billion euro (EAA, 2016). For Western-Europe, the IPCC predicts that the amount and intensity of precipitation will increase considerably in the coming decades, leading to more extreme events, and concludes that an efficient water regulation policy will be the most important challenge of the 21[st] century (IPCC, 2013).



## 1.2 The urban green

Part of the solution can come from a strategic use of rainwater in urban environments; this approach is known under different names such as 'Water Sensitive Urban Design (WSUD)'(Wong et al., 2013), 'Low Impact Development (LID)' (Dietz, 2007) and 'Sustainable Drainage Systems (SuDS)' (Ciria, 2013). One of their main goals is to try to use rainwater as efficiently as possible in the city itself. Among other measures this approach emphasizes the role of urban trees in the hydrological cycle. Urban trees are known to intercept rainwater, thereby creating a buffer for peak runoff during very intense rain events (Livesley et al., 2016; Xiao & McPherson, 2011). Urban built-up areas also benefit from trees during less intense events because of the reduced amount of rainwater that needs to be processed by the sewage system. Xiao & McPherson (2002) found that the trees in Santa Monica, California intercepted 1.6 % of annual precipitation, thereby saving $110,890 of the costs for flood control, or $3,60/tree. They further advise that planting more large, evergreen trees would increase the long-term benefits in runoff reduction benefit to be $47.3/tree in Lisbon. Most of the rainfall that trees intercept evaporates into the atmosphere, diminishing and redistributing the net rainfall that reaches the ground and is converted to surface runoff. Moreover, green spaces disrupt the impervious cover and allow rain water to infiltrate and contribute to the often depleted ground water tables under cities (Armson et al., 2013; Farrugia et al., 2013; Shields & Tague, 2014) .

Urban trees normally display a different behaviour than forest trees due to interactions with anthropogenic structures. Urban built-up affects wind orientation and rainfall, and creates microclimates which influence tree growth and well-being and hence rainfall interception (Pretzsch et al., 2017; Zipperer et al., 1997) . Asadian & Weiler (2009) found urban trees intercepting twice as much water as their forest counterparts, possibly due to the UHI effect, to the greater distance between trees (boundary layer effect) and to the open grown canopies. Studies done on natural forest interception thus cannot be easily translated to urban trees.

Several authors have attempted to quantify the contribution of urban trees on the total water balance. Some authors looked at large areas of urban cover and determined the water storage potential of the urban forest based on land cover derived maps (Gill et al., 2007; Haase, 2009; Verbeeck et al., 2013). Thereby they assign empirical storage capacity values to certain vegetation classes based on literature. These studies can give an accurate estimate of the total outflow on a catchment scale, but often fail to account for the smaller scale and the complex heterogeneity specific to an urban environment. Other authors looked at the influence of a single urban tree on the water balance (Asadian & Weiler, 2009; Guevara-Escobar et al., 2007; Véliz-chávez et al., 2014; Xiao et al., 2000a, 2000b). These studies generally accurately described the interception process for an isolated urban tree. However, due to the use of complex variables for the prediction of the interception process, the results are often difficult to extrapolate to larger areas and/or other tree species. Moreover, these studies are usually done in Mediterranean climates which have a distinct precipitation pattern with dry summers and wet winters. In comparison, temperate climates have a more evenly distributed rainfall pattern. Extrapolating results from previous studies is thus not straightforward.

## 1.3 The hydrological processes

Several hydrological processes interact with each other when gross precipitation falls on a tree (Fig. 1):





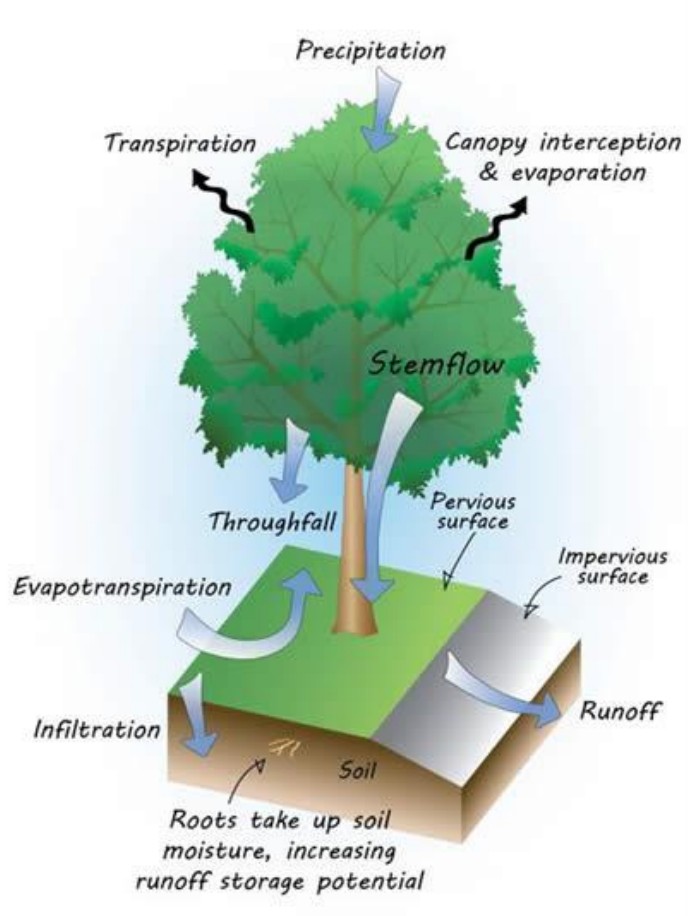

**Figure 1: Hydrological processes at work on the individual tree level (Image from: (EPA, 2015)).**

During the initial stage of the rain event, most water is intercepted by the tree canopy. Tree interception is defined as the process of precipitation falling on the tree surface where it is temporally stored. This water then either evaporates into the atmosphere, is absorbed by the leaves, flows down as stemflow or falls through/ drips off to the ground surface (Xiao et al.,2000a). The water that reaches the ground surface is called the net precipitation. A part of this water infiltrates in the ground, the other part runs off. The infiltration/runoff ratio depends on the surface- and soil properties. Part of the infiltrated water is taken up by the tree roots and is transpired back in the atmosphere through the leaves. The other part of infiltrated water replenishes the groundwater table. The total amount of water intercepted during an event is called the interception storage (mm). The main vegetation characteristic influencing the interception storage is the interception storage capacity (mm). This is the maximum amount of water the tree can hold for a given time. There is some confusion in literature regarding the exact definition of the interception storage capacity. This study utilizes the definition used by Xiao & McPherson (2016). They defined two types of interception storage capacity. The first is the surface saturation or minimum storage capacity, which is the amount of intercepted water that is needed on a vegetation unit for flow to begin. This water evaporates back into the atmosphere and does not contribute to throughfall. This type of storage is relatively independent of meteorological characteristics. Vegetation characteristics determining the minimum storage capacity are the canopy architecture, the leaf- and stem surface areas, the seasonal vegetation development and the tree's health condition (Asadian & Weiler, 2009; Véliz-



Chávez et al., 2014; Xiao et al., 2000b). The second type of storage is the detention- or maximum storage, this is the maximum amount of water that can temporally be stored on a vegetation unit. The maximum storage can temporarily exceed the minimum storage during very intense rainfall periods when the amount of water that falls through is smaller than the amount of water that is intercepted. However, once a threshold is reached or rain ceases, this extra amount of water quickly drips off until the minimum storage is reached again. This process is clearly observed by Keim et al. (2006) in rainfall simulator experiments on woody vegetation. They defined the two types of storage as the static- and dynamic storage respectively. In our study, we will use the surface saturation or minimum storage. Because this is the volume of water that never reaches the ground and does not contribute to runoff, it is of the most useful to hydrological modelers. The most convenient way of expressing this term is on the basis of the vertical projection area (VPA) in mm. The interception storage of an event can be larger than the interception storage capacity when intra-event evaporation or drip-off is present and the interception storage capacity is partially emptied and then filled again with new precipitation.

## 1.4 Measurement methods

Interception storage experiments can be conducted both in ex-situ and in in-situ conditions. Ex-situ experiments usually involve simulating rainfall events in a controlled environment. This allows to accurately determine the amount of rainfall intercepted and which vegetation- and meteorological characteristics are of most influence (Keim et al., 2006; Smets et al., 2018; Xiao & McPherson, 2016). Advantages of this method are that experiments are repeatable, that experiments can be designed to optimally determine the influencing variables and that many individual plants can be used. This method is usually used for smaller green elements like shrubs and grasses. Trees are impractical to transport to laboratories due to their above- and below ground biomass. Laboratory experiments have been done on tree branches (Keim et al., 2006; Xiao and McPherson, 2016), but upscaling to a whole tree level complicates the applicability of the results.

In-situ interception experiments usually involve the collection of rainwater above or besides the canopy and comparing this with rainwater collected under the canopy. The difference is the amount of intercepted water. An often used method is placing tipping buckets under the tree canopy (Asadian & Weiler, 2009; Link et al., 2004). This method only catches part of the throughfall. Throughfall however is usually not equally distributed under the canopy, which makes upscaling results to a whole tree canopy level difficult. Another in-situ measurement method is to collect all throughfall under a tree by constructing a V-catchment construction large enough to cover the canopy area (Véliz-Chávez et al., 2014; Xiao et al., 2000b). Throughfall and stemflow are usually collected in separate containers. A disadvantage of this method is that it is resource intensive and only a few individuals can be measured. Moreover, high wind speeds can cause lateral rain to be intercepted by the V-catchment and can confuse the measurements. However, because upscaling results from individual branches or leaves to the individual tree level remains difficult (Friesen et al., 2015), measurements on the whole tree scale are currently viewed as the most accurate method to quantify rainfall interception by solitary trees and are the preferred method in this research.

## 1.5 Interception models

The most used methods to calculate interception storage of a forest canopy are the model of Rutter (Rutter et al., 1971) and the analytical adaptation of his model by Gash (Gash &Morton, 1978). Their conceptual models include gross precipitation, crown storage, throughfall, stemflow and evaporation. These models serve as a starting point for most ulterior interception models (Muzylo et al., 2009). They are most commonly used on weekly/monthly temporal scales and a spatial scale of a forest stand. Rutter calculates the interception storage with a running water balance approach whereas Gash considers a wetting, saturation and drying phase to include the different water balance components. An important difference between the Gash and the Rutter model is that Gash considers rainfall events as discrete events. His model assumes an empty storage compartment at the start of each event and after reaching the saturation phase the amount of water intercepted is held constant and throughfall is assumed to start. Further, Gash & Morton (1978) treat throughfall as a factor in the water balance, while Rutter (1971) uses empirical relationships. Later, Gash refined his model to include open spaces in forests by including a canopy fraction cover (Gash et al., 1995). The canopy fraction cover enables the prediction of interception in open forest structures, and of the amount





of interception with a changing leaf cover. Van Dijk and Bruijnzeel (2001) later adapted the refined Gash model to include the leaf area index (LAI). They assumed a linear relationship between the LAI and the interception storage, thereby highlighting the importance of the leaf area in predicting the interception storage capacity. The first model to estimate interception storage on a single tree has been developed by Xiao (Xiao, 2000b). He adapted the Rutter model into a 3-dimensional physically-based

stochastic model to gain better understanding of the interception processes from a single leaf to the branch segment and then to the individual tree. He found the interception storage capacity to be the most important factor determining the amount of rain intercepted, followed by the LAI. The most influential meteorological factor for the interception storage was gross precipitation (Xiao, 2000b). The model provides a good tool to better understand the influence of the tree architecture and the detailed meteorological factors on the interception of a single tree in an urban environment. However, the intense model

parameterization makes it difficult for application.

Interception storage is the first part within a water balance simulation; estimating net rainfall available for infiltration, evapotranspiration and surface runoff. Therefore it is important to not only focus on interception models but also investigate the capacities of a water balance model to simulate interception storage. The Water and Energy Transfer between Soil Plants and Atmosphere simulator (WetSpa) allows a detailed modeling of the land surface processes (Wang et al., 1996). For this

study, the focus is on WetSpa's capacity for simulating the interception storage for a V-catchment set-up (Salvadore et al., 2015; Verbeiren et al., 2016; Wirion et al., 2017).

### 1.6 Research questions

Downscaling current hydrological interception models, which are mainly built for forested areas using a stand scale, to an individual urban tree might not reach satisfactory simulation results. Interception models developed on the individual tree scale are usually complex and require many variables that are difficult to measure in the field. This study evaluates different models for the simulation of the interception storage on urban trees while keeping model input limited to a few variables that are easy to measure.

The objectives of this study are:

- to evaluate the interception storage of two urban trees within Belgium (Western-European temperate climate).
- to evaluate how accurate different modeling frameworks describe the interception process on an individual tree level.
- to illustrate the importance of trees for the urban surface water balance, by simulating the changes due to the removal of trees on a university campus site.

### 2. Materials and methods

#### 2.1 The study area

The study area is located in Belgium. The climate of Belgium can be classified as Cfb Climate according to the Köppen climate classification (Kottek et al., 2006): a temperate oceanic climate with the coldest month averaging above 0°C, all months with

average temperatures below 22°C and at least four months with average temperatures above 10°C. Rainfall averages to 750 - 850 mm on a yearly basis and is fairly evenly distributed throughout the year. We hypothesize that in temperate climates a larger percentage of rainfall can be intercepted by urban trees than in Mediterranean climates where rainfall is usually restricted to the winter season.

#### 2.2 The selected tree species


Two deciduous trees were selected for this study: a Norway maple (*Acer platanoides L.*) and a small-leaved lime (*Tilia cordata Mill.*). Both trees are native in extensive parts of West- and East Europe and are introduced in large parts of the European





continent. They are popular street trees in urban environments due to their pollution removal abilities (Yang et al., 2015) and due to their growing rate at a young stage (Moser et al., 2015). The Norway maple is located at the VUB Campus Etterbeek (50°52' N, 4°41' E) in the capital region of Brussels. The small-leaved lime is found in 'Kasteelpark Arenberg' (50°52' N, 4°41' E). The Norway maple and small leaved lime were at least 8 m away from obstructions, minimizing possible influences

5    from nearby trees or buildings (Fig. 2).

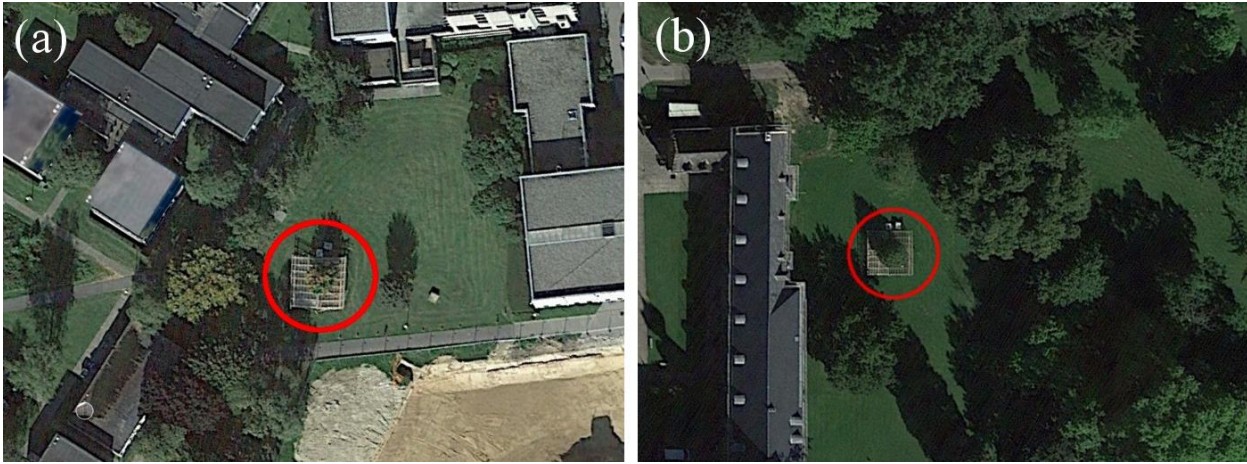

**Figure 2: Satellite images of the Norway maple (a) [01/10/2015] and small leaved lime (b) [25/08/2016] [images: Google Earth].**

### 2.3 The V-catchment design

A rainfall catchment was constructed under both trees to make throughfall measurements. The design of the construction was inspired by Xiao et al., 2000a. The skeleton was made out of *Pinus sylvestris* and covered with corrugated sheets (Super-Kristal, 450 g/m²) that intercepted rainfall and guided the water in a gutter that fed into a catchment container with a volume of 1 m³. The corrugated sheet was attached to the wooden skeleton with screws on the high sides of the waves. These screws

15   were then topped with a rubber sealing to avoid water leaking. Stemflow was collected by spiraling a half-open garden hose (Ø 2.5 cm) around the tree stem. This hose led to a separate stemflow container with a storage capacity of 26.75 L (Fig. 3).





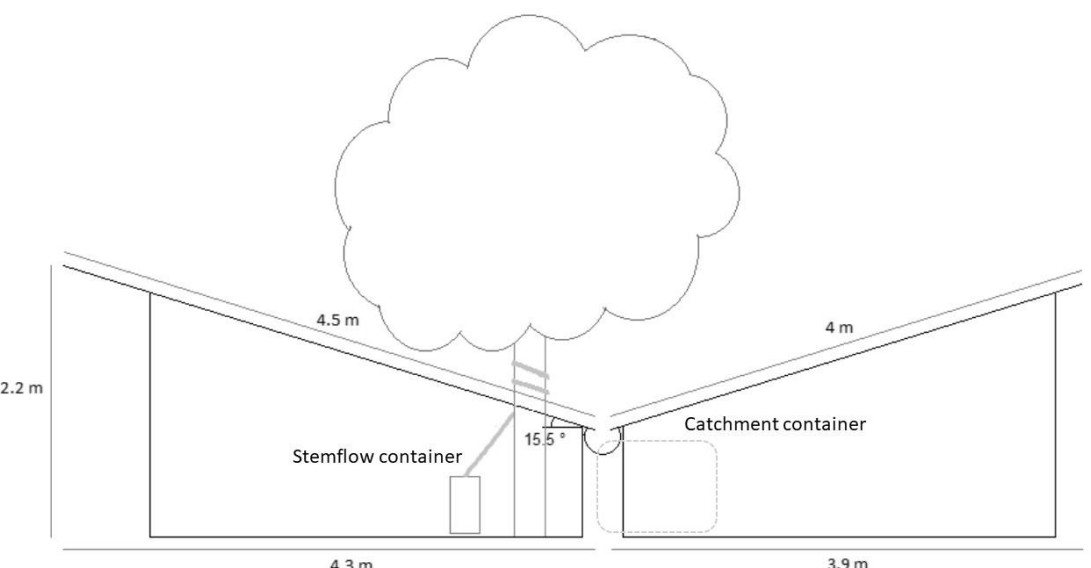

**Figure 3:Schematic representation of the V-catchment construction.**

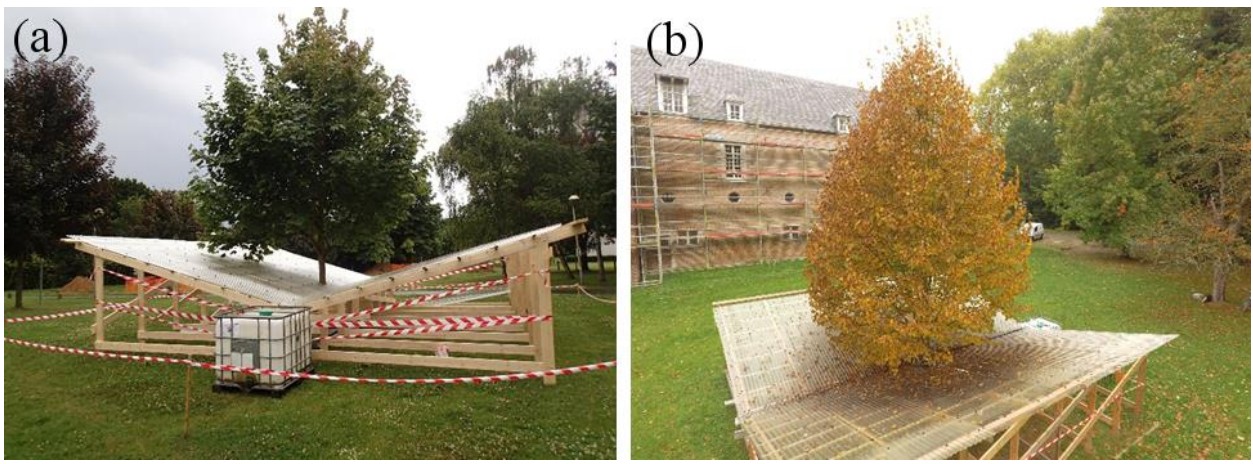

**Figure 4: Images of the  V-catchment under the Norway maple (a) [picture: 30 July 2015] and small leaved lime (b) [picture: 21 October 2016] .**

The catchment measured throughfall under the Norway maple from August 2015 until August 2016. Under the small-leaved lime, it was operational from September 2016 until August 2017. During winter, when no leaves were present on the trees and throughfall registration could be affected by frost and snow, no measurements were performed. Measurements reinitiated when the trees started to grow leaves again. Measurements were paused from 15 July 2015 until 23 March 2016 for the Norway maple and from 13 November 2016 until 14 Abril 2017 for the small leaved lime. Both trees were of similar dimensions and no substantial adaptation to the catchment had to be made. The V-catchment under the Norway maple had a total surface area of 66.4 m² and a vertical projection area of 65.04 m². The V-catchment under the small-leaved lime had a total surface area of 68.9 m² and a vertical projection area of 66.4 m².



Reference data from a meteorological station was used for the measurement of the gross precipitation. A rain event was defined as a rain volume record of minimum 0.1 mm, registered by the tipping bucket. In accordance with authors such as Asadian & Weiler (2009) and Staelens et al. (2006), rain events were separated by a dry gap of minimum four hours. For each rain event, characteristics such as duration, intensity and intra-event rainfall intermittency (IERI) were determined (Dunkerley, 2015).

For the period August 2015 until November 2015, a tipping bucket was installed on the top of a flat roof, approximately 20 m from the Norway maple. Additional meteorological data for the period August 2015 - November 2015 was gathered from a nearby weather station (RMI, Uccle). For the period March 2016 until August 2016, a meteo station was installed approximately 100 m from the Norway maple. Besides a tipping bucket (0.02 mm/ tip), this meteo station also included measurements of temperature, humidity, wind speed, wind orientation and solar radiation. These data had a temporal resolution
of 5 minutes. For the period September 2016 until August 2017, a commercially operated meteo station was used located approximately 1.8 km from the small-leaved lime. The precipitation data from the tipping bucket had a resolution of 0.01 mm and a time resolution of 15 min. This station also provided several other meteorological measurements such as temperature, humidity and wind speed (Table 1). The difference in time resolution between the reference stations did not affect measurement results in a significant way because of the above-mentioned definition of a rain event.

**Table 1: The meteorological stations.**

| Norway maple | Period | Ref station | Distance to tree | Rainfall resolution | Time resolution | Other measurements |
|---|---|---|---|---|---|---|
| 1st Measurement period (n=13) | 14/08/2015 - 14/10/2015 | 1 | 20 m | 0.1 mm | 1 sec | No |
| 2nd measurement period (n=26) | 24/03/2016 - 29/07/2016 | 2 | 100 m | 0.02 mm | 5 min | Yes |
| **Small leaved lime** | | | | | | |
| 1st measurement period (n=8) | 29/09/2016 - 12/11/2016 | 3 | 1.8 km | 0.01 mm | 15 min | Yes |
| 2nd measurement period (n=17) | 15/04/2017 - 18/08/2017 | 3 | 1.8 km | 0.01 mm | 15 min | Yes |

## 2.4 The Sensors

The water level in both the catchment- and stemflow container was monitored by pressure sensors (Mini Diver Dl501, Schlumberger Water Services). The sensors are calibrated by the manufacturer and have an accuracy of 5 mm. They store up to 24 000 measurements and are programmed to measure in specific time intervals. For this study, the sensors were set to measure every 30 seconds. Using this time interval, the sensors register data for 8.33 days before the memory is full. An identical sensor was installed on the same height under the tree to act as a barometer. This way the measurements in the
containers are compensated for atmospheric pressure before translating them to water column height (cmH20).

The LAI was measured periodically with the SunScan system (Type SS1-COM-R4). This system uses photodiodes to measure global and diffuse radiation either as photosynthetically active radiation (PAR) ($mmol\ m^{-2}\ s^{-1}$) or energy ($W\ m^{-2}$). By measuring the incoming radiation in eight compass directions under the tree, an approximation of the energy received by the ground surface under the tree can be made. By comparing these values with a reference sensor that is placed outside the tree canopy,
an estimation of the energy absorbed and reflected by the tree is made. A conversion to LAI is done using an equation based



on the Beer-Lambert Law. For a full description of the methodology and validation of this procedure we refer to Wirion et al. (2017) .

## 2.5 The accuracy of the measurements of the water balance components
### 2.5.1 The accuracy of the volumes stored in the containers

The pressure divers had an accuracy of 5 mm $H_2O$. To calculate the water stored in the container after a rainfall event, two diver readings had to be made (initial- and end value). The resulting accuracy will thus be the sum of the two reading accuracies (10 mm). As the vertical projection area of the container was 1.12 m², 10 mm of water level rise translates to a volume difference of 11.2 liter. Because the catchment container had rounded corners, the water level was always kept between 200 and 700 liters, meaning that no correction factor was needed to calculate throughfall. This meant that the maximum capacity of the storage container was reached after an event of approximately 7.5 mm (493 liter) , without taking the storage of the tree into account. Because of the storage capacity of the tree, the actual rain event size needed to fill the storage container was slightly higher. The stemflow container was cylindrical shaped with a vertical projection area of 0.0625 m². One cm rise in water level equals a volume difference of 0.3125 liters. The total volume of the stemflow container was 26.75 liters, which was large enough to capture stemflow of a rain event of any realistic size.

### 2.5.2 The detention storage of the V-catchment

After a rainfall event, a certain amount of water is retained on the corrugated sheets of the V-catchment and evaporates into the atmosphere again. To quantify this amount, a detention storage measurement was performed. A corrugated sheet of 0.209 m² was positioned at the same angle as the construction (15.5°) and sprayed with water until droplets started flowing of the bottom edge. The remaining water was then collected after one minute with highly absorbing tissues. The difference in tissue weight before and after absorbing water delivered the amount of water retained on the corrugated sheet. The balance had a precision of 1 g. This process was repeated 10 times, which led to an average water retention on the corrugated sheet of 16 g, equivalent to 0.077 mm/m².

To calculate the detention storage of the rain gutter, a spare piece of gutter measuring 3.27 m was hanged on the same angle as the rain gutter on the construction (1cm descend/m = 0.57°/m). A known volume of water (2000 mL) was poured into the gutter and was collected again on the lower end. Three repetitions were made and an average of 1974 mL was collected again. This means that per meter of rain gutter, 8 mL of water is retained. For the whole construction the gutter measured 8.25 m, which accumulates to a total of 66 mL of water retained.

These two sources of inaccuracy, the sensor inaccuracy and detention storage, had a smaller effect for larger rainfall events. To quantify this effect, a sensitivity analysis was done. For both the Norway maple and the small leaved lime, rain events of several sizes were analyzed. For every rain event size, the decrease in percentage interception storage with 10 mm water level rise of the container was analyzed. Results are slightly different for both trees because the vertical projection area of the trees differ slightly. A sensitivity analysis of the detention storage was done by simulating 1 liter of variation in the detention storage (corrugated sheet- and gutter detention) for rain events of different sizes. No sensitivity analysis of the stemflow was done because very few events displayed significant stemflow.



## 2.6 The data processing

The following paragraph describes how to calculate throughfall, interception storage and stemflow data from the raw data received by the pressure sensors and tipping buckets:

The water balance for a rainfall event states:

**(1)**

$$P_g = I + T_f + S_t$$

where $P_g$ is the gross precipitation, $I$ is the interception storage of the tree, $T_f$ is the throughfall under the tree and $S_t$ is the stemflow of the tree.

$P_g$ (mm) is recorded by a pluviometer close to the V-catchment and extrapolated to the vertical projection area of the construction to calculate the total water volume that falls on catchment surface. Part of the $P_g$ falls directly on the catchment and is guided to the catchment container ($P_{g\ \text{free}}$). Another part of $P_g$ falls on the tree ($P_{g\ \text{tree}}$). A small part of this water is free

throughfall that never comes in contact with the tree. The majority of water however is intercepted by the tree's leaf- and stem surfaces. Once the interception storage capacity of the tree is filled, throughfall will occur. Water that flows downwards from the stem gets collected in a separate stemflow container. The interception storage of the tree is then the difference between on the one hand the gross precipitation fallen on the whole construction during the event and on the other hand the sum of the precipitation fallen on the free construction, the throughfall and stemflow readings. For the V-catchment the water balance

reads:

**(2)**

$$P_g = P_{g\ \text{free}} + P_{g\ \text{tree}}$$

**(3)**

$$P_g = P_{g\ \text{free}} + I + T_f + S_t$$

After including the vertical projection areas, the water level readings of the pressure sensors and rearranging Eq. 3 becomes:

**(4)**

$$I = \left(P_g * \text{VPA}_{\text{constr}}\right) - \left(\left(P_{g\ \text{free}} * \text{VPA}_{\text{free}}\right) + (\Delta H_{\text{cont}} * \text{VPA}_{\text{cont}}) + (\Delta H_{\text{St}} * \text{VPA}_{\text{St}})\right)$$

$\text{VPA}_{\text{constr}}$, $\text{VPA}_{\text{free}}$, $\text{VPA}_{\text{cont}}$, $\text{VPA}_{\text{st}}$ are the vertical projection areas of the whole V-catchment construction, the part of the

construction not covered by the tree, the catchment container and the stemflow container respectively. $\Delta H_{\text{cont}}$ and $\Delta H_{\text{St}}$ are the height differences recorded after a rain event in the catchment- and in the stemflow container.

Not taking the tree interception into account, the catchment container reached its maximum capacity (493 L) when a rain event of 7.5 mm occurred. To take into account larger rainfall events, we followed the procedure explained below:





Each rain event that filled the catchment container was divided in two parts: the first part lasts until the container is filled and the second part starts when the container is full and all additional rain overflows to the ground. If the amount of rain fallen until the moment the container filled was larger than the interception storage capacity of the tree, it was assumed that the interception storage capacity was reached and all additional water would be converted to throughfall. Otherwise the event was discarded. The throughfall values of the first and second part of the event are then summed and compared to the interception storage values of the first part of the event. The stemflow container never filled completely and was analyzed on a whole event basis.

The interception storage capacity (S; mm) was calculated with an empirical equation based on LAI (-) (Gómez et al., 2001):

**(5)**

$$S = 1.184 + 0.490 \, \text{LAI} \quad R^2 = 0.76$$

For comparison, we also used the method developed by Leyton et al. (1967) to calculate the interception storage capacity. A line of unit slope, minus the stemflow percentage, is drawn on a gross precipitation vs throughfall plot through those rain events where evaporation is assumed minimal. Only events where $P_g$ is large enough to fill the storage capacity are used. The determination of the amount of precipitation to reach canopy saturation is subjective and based on the recognition of an inflection point on the graph. To make recognition of this inflection point easier, the residuals of the $P_g$ vs $T_f$ regression were plotted against the $P_g$. The inflection point is recognized by a sudden change in the variability of residual values. The interception storage capacity is then found by the negative intercept of this upper envelope line with the y-axis. The upper envelope represents events with minimal evaporation. The points left of the inflection point represent events where rainfall was insufficient to fill the canopy storage completely. The parameter of the gross precipitation of a regression through the lower envelope of these points represents the free throughfall coefficient. An example of this method from (Sadeghi et al., 2015) is shown in Fig. 5:



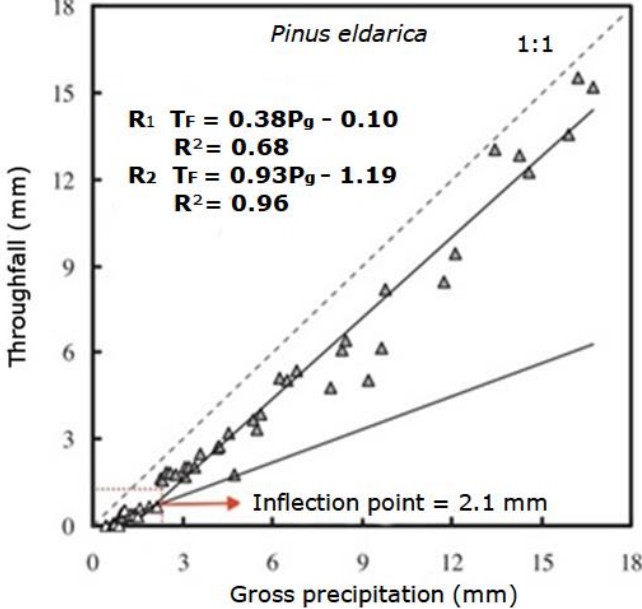

**Figure 5: Example of the Leyton method from (Sadgehi, 2015). The lower- and upper envelopes are represented by equations R1 and R2 respectively. In this example the interception storage capacity of a Pinus Eldarica is 1.19 mm and the free throughfall coefficient is 0.38. With $T_f$ being the Throughfall and $P_g$ the Gross precipitation.**

## 2.8 Modeling and model comparisons

The measured interception storage will be compared to different simulation approaches: the Gash, Rutter, WetSpa and classical regression analysis (Table 2). The equations for the different simulations of the interception storage can be found in appendix A. The Gash model -as well as the regression method- consider separate rainfall events and the interception storage capacity is assumed to be completely emptied before each event. The continuous simulations with Rutter and WetSpa are performed at the same timestep as the rainfall measurements (see §2.3). This approach enables an emptying of the storage by evaporation during the event. The equations of Gash and Rutter have been developed for a forest stand, whereas the WetSpa model is adapted to a single tree/ V-catchment (Salvadore et al., 2015). For all models we estimate the interception storage capacity with the measured LAI (Eq. 5, Gómez et al., 2001). Gash, Rutter and WetSpa empty the storage via evaporation from the leaves based on the potential evapotranspiration estimated with the Penman-Monteith equation (Monteith, 1965). Gash estimates free throughfall with the gap fraction (Gash and Morton, 1978; Leyton et al., 1967; Rutter et al., 1971; Xiao et al., 1998). Further drip-off from leaves is estimated using the simplification of Gash et al. (1999), in order to avoid empirical parameterization (Rutter et al., 1971).

As an addition to the simulations we also use a linear regression analysis. For urban planners or policy makers such a method can be more straightforward and thus more appealing to use. To select the most significant predictor variables for the regression, a PCA analysis was performed, resulting in 3-5 principal components. Based on this PCA analysis, a clustering of the variables was made. A clustering algorithm was used that iteratively splits the variables in smaller clusters according to the principal components of the clusters. Splitting occurs on the cluster with the largest second eigenvalue. Once split, a correlation analysis is performed and the variables are reassigned to the cluster they are most correlated with. The algorithm



stops when the second eigenvalue of all the clusters is smaller than one. Initially, 5 clusters were found. All the variables in each cluster were then fitted to the interception storage (mm). The two clusters with the lowest predictive performance were eliminated. The variables inside the 3 remaining clusters were examined separately. Based on their individual predictive performance and expert knowledge, one variable was selected from each cluster and used as an independent variable in a multiple linear regression of the Norway maple dataset. Before fitting, variables were normalized in the range [-1,1] to make the parameter estimates more meaningful. The dataset of the small leaved lime was used to validate the regression equation.

**Table 2: Characteristics of the different simulation approaches.**

| | Time step | Spatial extent | Interception storage capacity | Drip-off | Evaporation during event | Hydrological processes simulated |
|---|---|---|---|---|---|---|
| **Gash** | Discrete | Forest stand | LAI (eq. 5) | Yes | Yes | Interception, throughfall |
| **Rutter** | Continuous | Forest stand | LAI (eq. 5) | Yes | Yes | Interception, throughfall, evaporation |
| **WetSpa** | Continuous | Single tree | LAI (eq. 5) | No | Yes | Interception, throughfall, evapotranspiration, infiltration, depression loss, runoff, … |
| **Regression** | Discrete | Single tree | / | No | No | Interception |

### 2.9 The scenario analysis

To illustrate the effects of the interception storage in an urban environment, we simulated the surface water balance of the university campus of the Vrije Universiteit Brussel (VUB) before and after the construction of new student houses. The construction area covers 16692 m². Before the construction started in July 2015, trees covered 15 % of the area, grass 45 %, bare soil 3% and urban materials 37 % (Fig. 6c). After the construction works, trees only cover 2 %, grass 4 %, bare soil 33 % and urban materials 61 % (Fig. 6d).

To analyze the effects of land cover change on the net rainfall and thus on the surface water balance we use a calibrated WetSpa model (Wirion et al., 2017). Wirion et al., 2017 calibrated the WetSpa model for the Watermaelbeek catchment for the year 2015. We use the same meteorological input (709 mm of rain in 2015).

The LAI of trees before the construction started has been measured in 2015. On average the trees had a LAI of 0.52 in low leaf conditions (measured April 10, 2015) and a LAI of 2.95 in maximum leaf conditions (measured June 4, 16 and July 6, 2015). The measured LAI values are used to estimate the interception storage of the trees before the construction starts. For the interception occurring on grass and bare soil constant values, retrieved from remote sensing data, are used (Wirion et al., 2017). We assume that urban surfaces do not intercept rainfall.

We analyze two scenarios to evaluate the effect of different landscape design scenarios on the university campus site: (1) an urban scenario, where all bare soil land cover is sealed/turned impervious once the construction works are finished and (2) a natural scenario, where all bare soil land cover after construction is covered by urban trees. For the second scenario we attribute the characteristics of the Norway maple tree to the newly covered tree areas. Further, we assume soil characteristics to be similar to clay as it has become highly compacted due to anthropogenic activities.



**Figure 6: The university campus before and after the construction of new student houses. 1) Orthophotos before [2012] (a) and after [2017] (b) the construction works. 2) land cover map before (c) and after (d) construction works; dark green: trees, green: grass, rose: urban, beige: bare soil.**



## 3. Results

### 3.1 Descriptive statistics

The vegetative characteristics of both trees are described in Table 3

5 **Table 3: Vegetation characteristics of the Norway maple and small leaved lime.**

|  | Norway maple | Small leaved lime |
|---|---|---|
| Diameter (m) | 8.92 | 8.79 |
| Diameter at breast height (cm) | 47 | 46 |
| Crown diameter (m) | 5.95 | 6.35 |
| Crown height (m) | 7.39 | 7.09 |
| Crown shape | Oval | Oval |
| Vertical projection area (m²) | 27.83 | 32.30 |
| Leaf area (cm²) | 90.36 +- 43.76 | 41.41 +- 17.63 |
| Average branch angle (°) | 38.15 +-17.54 | 41.29 +- 17.83 |

Both trees are similar in dimensions and characteristics. A noticeable difference is the average leaf area of the Norway maple, being twice as large as the leaf area of the small leaved lime. The change in LAI of both trees can be seen in Fig. 7. Both trees are deciduous. For both trees we measured the LAI on 7 moments during the season, to cover low, medium and full leave conditions. We then use a linear interpolation between each measurement to assign an LAI value for each rain event (Fig. 7). The LAI of the small leaved lime (LAI = 4.8) in full leave conditions is higher than the LAI of the Norway maple (LAI = 3.6). In minimum leaf conditions the LAI is lower for the small leaved lime (LAI = 0.5) than for the Norway maple (LAI = 0.58). The changes in LAI throughout the season are thus more important for the small leaved lime.





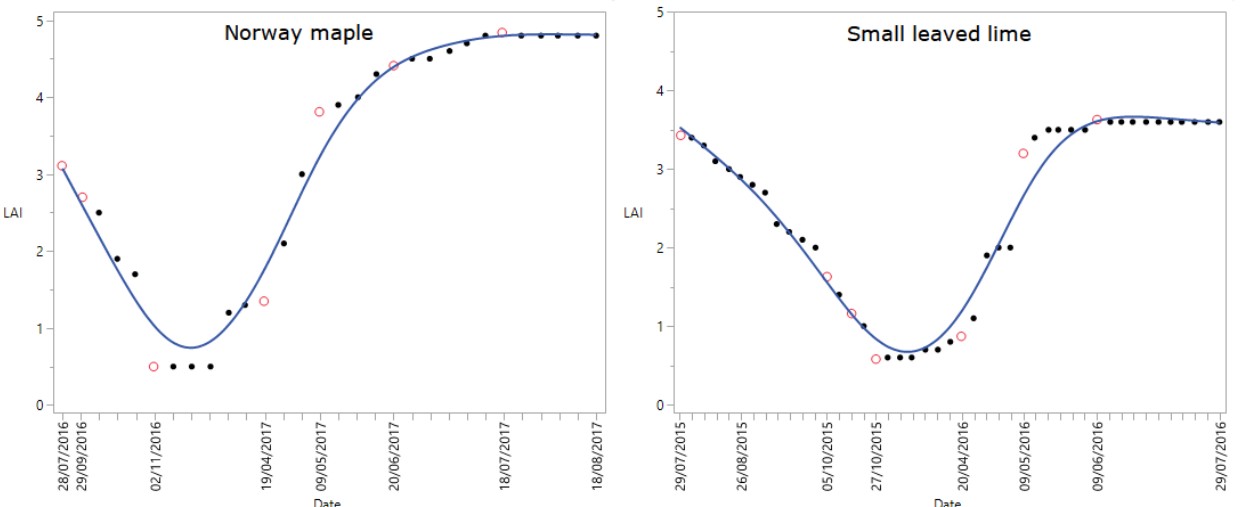

**Figure 7: The variation of the LAI of the Norway maple (left) and the small leaved lime (right). Hollow red dots indicate the dates of the measurements.**

Histograms of the precipitation, the event duration, the intensity , the intra rain event intermittency (IERI) (-) and the wind speed (m/s) are shown in Fig. 8. IERI is calculated by dividing the time it didn't rain during an event by the total event time. An IERI value close to zero means that it rained almost continuously during an event.





**Figure 8: Histograms with precipitation event characteristics for the Norway maple (left) and small leaved lime (right).**



Rain events on both sites are predominantly low in precipitation amount, duration and intensity. For the Norway maple site, an average event contained 3.7 mm of rain and lasted 325 min with an intensity of 1.3 mm/h. An average event on the small leaved lime site contained 4.7 mm of rain and lasted 392 min with an intensity of 0.9 mm/h. These intensities are lower in comparison to Mediterranean climates where average intensity > 2 mm/h is common (Pereira et al., 2009). The intra rain event intermittency (IERI) shows more equally distributed values across its spectrum, meaning rain events exhibit different lengths of intra-event dry periods. Average wind speeds are low for the Norway maple site but high for the small-leaved lime site. This is probably due to the complex wind patterns typically present in urban environments and to the more exposed location of the weather station of the small leaved lime. The values of these rain event characteristics are typical for temperate climates.

### 3.2 Event Summary

The gross precipitation of each event was divided in throughfall, interception storage and stemflow (Table 4).

**Table 4: Event summary of the precipitation events of both trees.**

| | | | Throughfall | | Interception storage | | Stemflow | |
|---|---|---|---|---|---|---|---|---|
| **Northern maple** | Events (#) | $P_g$ (mm) | Total (mm) | Percent (%) | Total (mm) | Percent (%) | Total (mm) | Percent (%) |
| Total | 39 | 143,71 | 88,03 | 61,26% | 55,61 | 38,70% | 0,19 | 0,13% |
| Events < 5 mm | 29 | 63,47 | 36,41 | 57,37% | 27,04 | 42,60% | 0,01 | 0,02% |
| Events 5-10 | 7 | 46,98 | 27,96 | 59,51% | 19 | 40,44% | 0,12 | 0,26% |
| Events 10-20 | 3 | 33,26 | 23,66 | 71,14% | 9,57 | 28,77% | 0,06 | 0,18% |
| M. period 1 (mm) | 13 | 41,1 | 21,41 | 52,09% | 19,69 | 47,91% | 0,07 | 0,17% |
| M. period 2 (mm) | 26 | 102,61 | 66,62 | 64,93% | 35,92 | 35,01% | 0,12 | 0,12% |
| | | | | | | | | |
| **Small leaved lime** | Events (#) | $P_g$ (mm) | Total (mm) | % | Total (mm) | % | Total (mm) | % |
| Total | 25 | 117,31 | 70,35 | 59,97% | 44,12 | 37,61% | 0,11 | 0,09% |
| Events < 5 mm | 17 | 37,52 | 18,3 | 48,77% | 16,43 | 43,79% | 0 | 0,00% |
| Events 5-10 | 4 | 27,9 | 12,05 | 43,19% | 15,81 | 56,67% | 0,03 | 0,11% |
| Events 10-20 | 4 | 51,89 | 40 | 77,09% | 11,88 | 22,89% | 0,08 | 0,15% |
| M. period 1 (mm) | 8 | 26,89 | 21,52 | 80,03% | 5,35 | 19,90% | 0 | 0,00% |
| M. period 2 (mm) | 17 | 90,42 | 48,83 | 54,00% | 38,77 | 42,88% | 0,11 | 0,12% |

Taking all events into consideration, both trees are very similar in behavior. Both trees intercept 38 % of the rain and stemflow is negligible for both trees. The largest difference is found in the events between 5 and 10 mm, where the Norway maple intercepts 40% of the rainfall while the small leaved lime intercepts 57 %. Another noticeable difference is seen when we compare the measurement periods. In the first measurement period the Norway maple intercepts significantly more (47.9%)



than the small leaved lime (19.9%). Interception storage in the second measurement period is more similar between both trees with 35 % of rain intercepted by the Norway maple and 42.9 % by the small leaved lime.

5    **3.3 The sensitivity analysis**

Figure 9a shows the uncertainty of the interception estimates associated with rain event size while Fig. 9b shows the influence of the detention storage on event size with a one liter variation in detention storage. The equations can be found in appendix B.

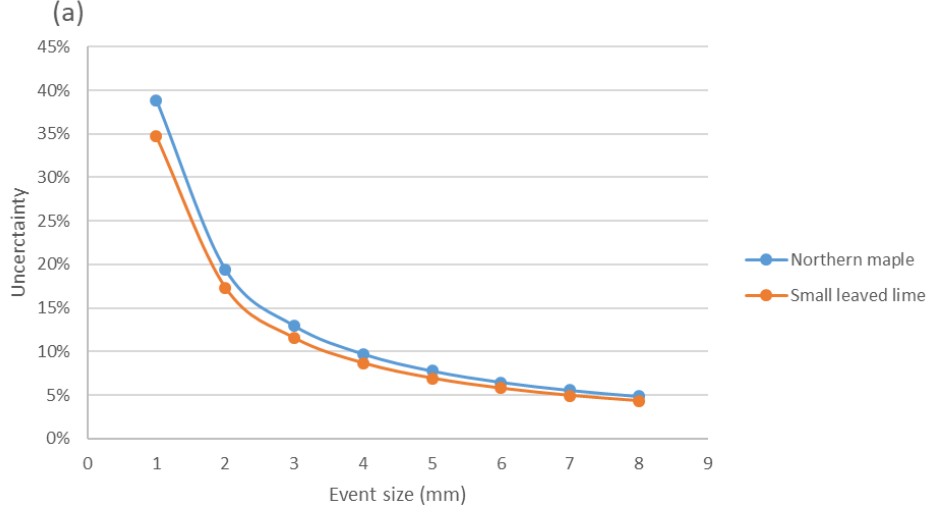

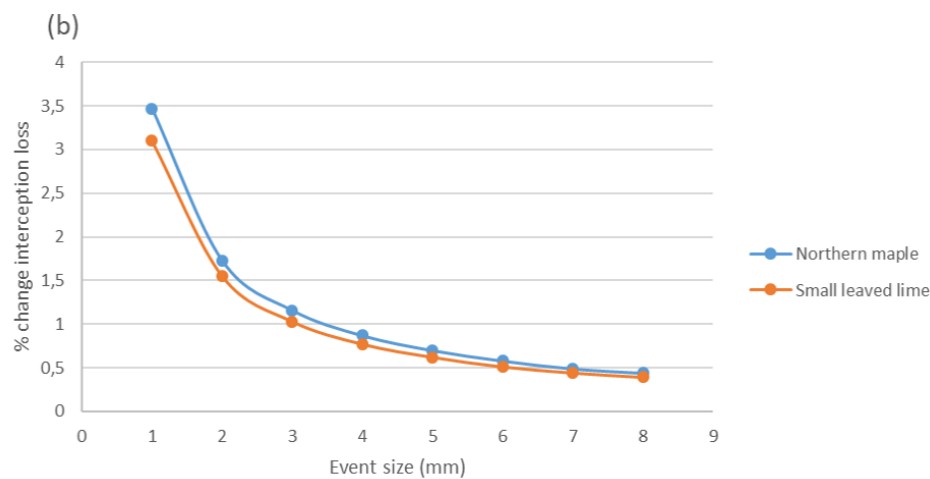

**Figure 9: Analysis of (a) the uncertainty in interception storage estimates with increasing event size and (b) Influence of the detention storage on interception storage estimates with increasing event size with a one liter variation in detention storage.**





As can be seen from Fig. 9a, for small events of 1 mm, uncertainty is quite high, being 38.8 % for the Norway maple and 34.7 % for the small leaved lime. This uncertainty drops fast however and for events with 2 mm size, uncertainty for both trees is beneath 20 %. For events larger than 4 mm, uncertainty drops below 10 % and for events larger than 7 mm, uncertainty is around 5 % .

5    The sensivity analysis of the detention storage yields far smaller values. For events of size 1 mm, a change of one liter in detention storage only changes the interception storage around 3 %. For events larger than 3 mm, this change in interception storage drops below 1 %. It is clear that the dominant error is the error associated with the pressure divers. Due to the high uncertainties associated with small events we decided to exclude events < 1 mm from the analysis.

### 3.4 Interception storage capacity and free throughfall coefficient estimation

Figure 10a and 10b show the determination of the canopy storage capacity for the Norway maple and small leaved lime for the whole measurement period with the method of Leyton et al. (1967). For the small leaved lime, two points right of the inflection point were chosen to be included in the lower envelope. The Leyton method allows for this to a certain degree due to the inherent subjectivity in the determination of the inflection point.

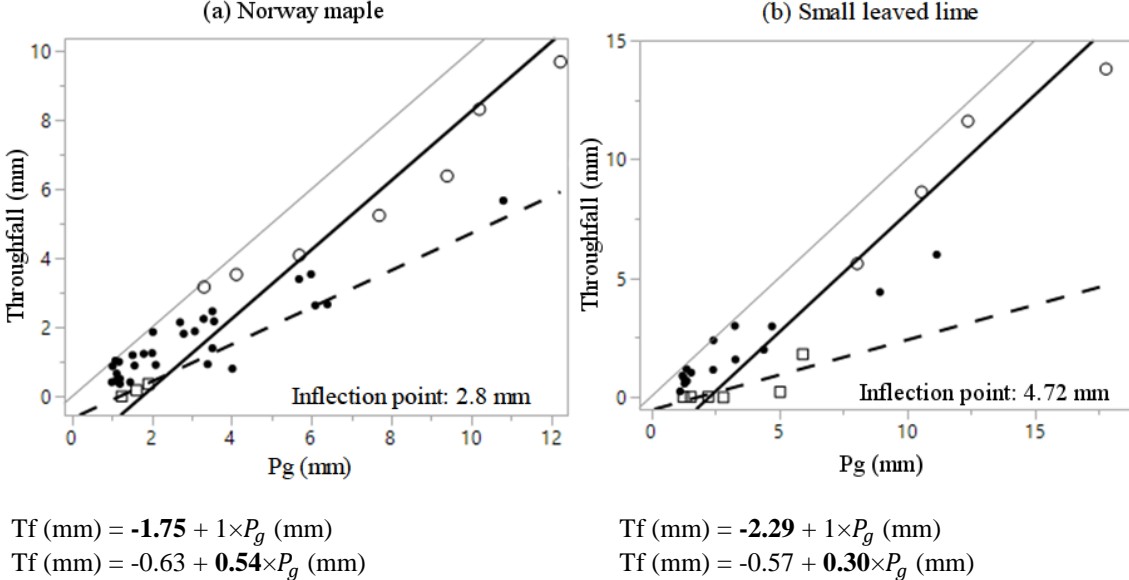

$$\text{Tf (mm)} = \mathbf{-1.75} + 1 \times P_g \text{ (mm)}$$
$$\text{Tf (mm)} = -0.63 + \mathbf{0.54} \times P_g \text{ (mm)}$$

$$\text{Tf (mm)} = \mathbf{-2.29} + 1 \times P_g \text{ (mm)}$$
$$\text{Tf (mm)} = -0.57 + \mathbf{0.30} \times P_g \text{ (mm)}$$

**Figure 10: Leyton regression analysis of (a) the Norway maple (n=39) and (b) the small leaved lime (n=25). Hollow dots are the points used to draw the upper envelope line; hollow squares are the points used to draw the lower envelope line. The points not used to draw the envelopes are depicted in black.**





**Table 5: Interception storage capacity (S; mm) and free throughfall coefficient (p; dimensionless) as calculated by the Leyton-method and the method of Gómez for all datasets.**

| Norway Maple | Period | S (Leyton) | S (Gómez) | p (Leyton) |
|---|---|---|---|---|
| All year (n=39) | 14/08/2015 - 29/07/2016 | 1.75 | 2.47 | 0.54 |
| 1st measurement period (n=13) | 14/08/2015 - 14/10/2015 | 1.71 | 2.40 | 0.6 |
| 2nd measurement period (n=26) | 24/03/2016 - 29/07/2016 | 1.85 | 2.50 | 0.32 |
| **Small leaved lime** | | **S (Leyton)** | **S (Gómez)** | **p (Leyton)** |
| All year (n=25) | 29/09/2016 - 18/08/2017 | 2.29 | 2.62 | 0.30 |
| 1st measurement period (n=8) | 29/09/2016 - 12/11/2016 | 0.38 | 1.90 | 0.47 |
| 2nd measurement period (n=17) | 15/04/2017 - 18/08/2017 | 2.54 | 3.14 | 0.25 |

Considering the whole year, interception storage capacities (S) for the small leaved lime are higher than for the Norway maple as expected when we consider the LAI measurements. Further we separate the measurements into measurement period 1 and 2, based on our measurement campaign. The timespan these periods cover are shown in Table 5. In general measurement period 1 and 2 cover both low- and high leave coverage. Therefore differences in S are not so outstanding. However in the period where trees lose their leaves S seems to be a bit lower than in the period where trees gain new leaves. The differences are higher for the small leaved lime which is also what we expect from LAI measurements. There is quite a discrepancy between the Leyton method and the method of Gómez (Eq. 5). The trends are similar, however the absolute values differ significantly. Both methods are within the range of expected interception storage capacity values found in literature and ranging between 0.2 and– 3.58 mm (André et al., 2008; Aston, 1979; Breuer et al., 2003; Gash & Morton, 1978; Gómez et al., 2001; Liu & Smedt, 2004; Valente et al., 1997; Xiao et al., 2000a). In their paper, Gómez et al. (2001) acknowledge a slight overestimation of the interception storage capacity whereas Gash and Morton, 1978 recognize a slight underestimation using the method of Leyton. For the small leaved lime tree, measurements in autumn are limited and thus the estimated interception storage capacity is unrealistically low. Further the values are in general a bit lower than the values estimated based on LAI. The Leyton method is a subjective method that also depends on the quality of the measurements (Gash & Morton, 1978). This makes the method of Leyton more difficult to apply than the method of Gómez which we therefore decided to use throughout this paper. The free throughfall coefficients for both trees are higher in the 1st measurement period (loosing leaves) than in the datasets of the second measurement period (gaining leaves). Further the gap fraction for the small leaved lime is smaller than for the Norway maple which corresponds to the LAI measurements. The values are comparable to literature where values are found between 0.1 - 0.86 for the gap fraction (Aston, 1979; Gash & Morton, 1978; Valente et al., 1997; Véliz-chávez et al., 2014; Xiao et al., 2000a).





### 3.5 The regression analysis

The three variables selected based on the clustering algorithm and expert knowledge are the gross precipitation (mm), the duration of the event (min) and the LAI (-). The results of the clustering algorithm can be consulted in appendix C. The resulting regression equation reads:

(6)

$$I\,(\text{mm}) = 3.12 + 1.37\left(\frac{D\,(\text{min}) - 620}{610}\right) + 1.47\left(\frac{P_{\text{g}}(\text{mm}) - 9.39}{8.39}\right) + 0.62\left(\frac{LAI - 2.65}{2.15}\right)$$

Where D stands for the rain event duration in minutes. All three independent the variables significantly contribute to the model (P value < 0.01). By normalizing the variables to fall in to a range of [-1,1], their relative contributions to the interception storage become apparent when looking at their regression coefficients. The gross precipitation has the largest influence on the interception storage, closely followed by the duration of the event. LAI also significantly influences the amount of water intercepted but its contribution is less important than the two above mentioned variables.

The R² and R²adj of the regression are 0.81 and 0.80 respectively with a root mean square error ($E_{RMS}$) of 0.52 mm. Only the events of the Norway maple (N=39) were used (Fig. 11).

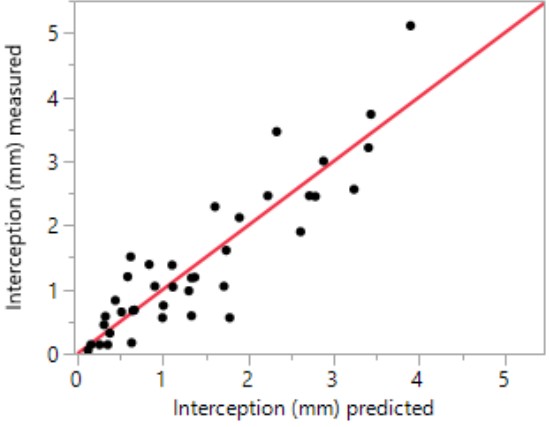

**Figure 11: Measured vs predicted interception using the regression analysis.**

### 3.6 The simulations

The V-catchment measurements include free throughfall, drip-off and stemflow. As our stemflow measurements are very low ( < 0.2 %) we exclude stemflow from our simulations. In general all simulation methods perform similarly and predict an interception storage close to the measurements (I = +- 40 % of rainfall) (Fig. 12). For bigger rainfall events (Pg > 10mm) the measurements indicate an average interception of 25% of rainfall whereas for smaller rainfall events (Pg < 10mm) we measure an average interception rate of 47 % of rainfall.





Table 6 shows that the method of Gash is most stable throughout the 2 trees and seasons (R² = 60% +- 2%, $E_{RMS}$ = 0.85 +- 0.2 mm). All other methods show more variation. The regression method for instance performs very well for the calibration tree (Norway maple, R² = 81%, $E_{RMS}$ = 0.52mm) whereas an important reduction in performance can be seen for the validation tree (small leaved lime, R² = 53%, $E_{RMS}$ = 1.08mm). It is thus difficult to use the linear regression equation of the Norway maple tree for other trees even with similar characteristics. It is also unclear if it should be used for different climate conditions as the performance varies a lot between the loosing leaves period (August – November) and the gaining leaves period (March – August).

As in the paper of Véliz-Chávez et al. (2014) the performance of the Rutter model is below the performance of Gash. Similarly, we also observe an underestimation of the interception storage for higher rainfall events ($P_g$ > 10 mm) with the method of Rutter (Fig. 12). WetSpa performs best for rainfall events $P_g$ > 10 mm and the regression method performs worst (Table 6 , Fig. 12, Fig. 13). For small events ($P_g$ < 10 mm) all simulations overestimate the interception storage. The regression method followed by Gash perform best for events $P_g$ < 10 mm.

In Fig. 13 we can observe that all models simulate lower interception storage than measured for the bigger interception events (> tree storage capacity) and higher interception storage than measured for smaller interception events ( < tree storage capacity). For small interception events the storage capacity is not filled in our simulations, and trees intercept all rainfall water. However, due to wind speed and direction, rain inclination angle, leaf zenith angle and other meteorological and tree architectural parameters not all rainfall water is intercepted even for small events. Even though we use the higher estimate for storage capacity (§cf paragraph 3.4), our models still underestimate the interception storage for bigger interception events. We assume that the emptying of the storage via evaporation from leaves is underestimated in our simulations.

With regards to the correlation values, WetSpa performs better than Rutter but worse than Gash. For the small leaved lime tree and for rainfall events where $P_g$ > 10 mm , WetSpa shows the best performance. This makes us confident to use WetSpa for estimating the net rainfall in urban environments.

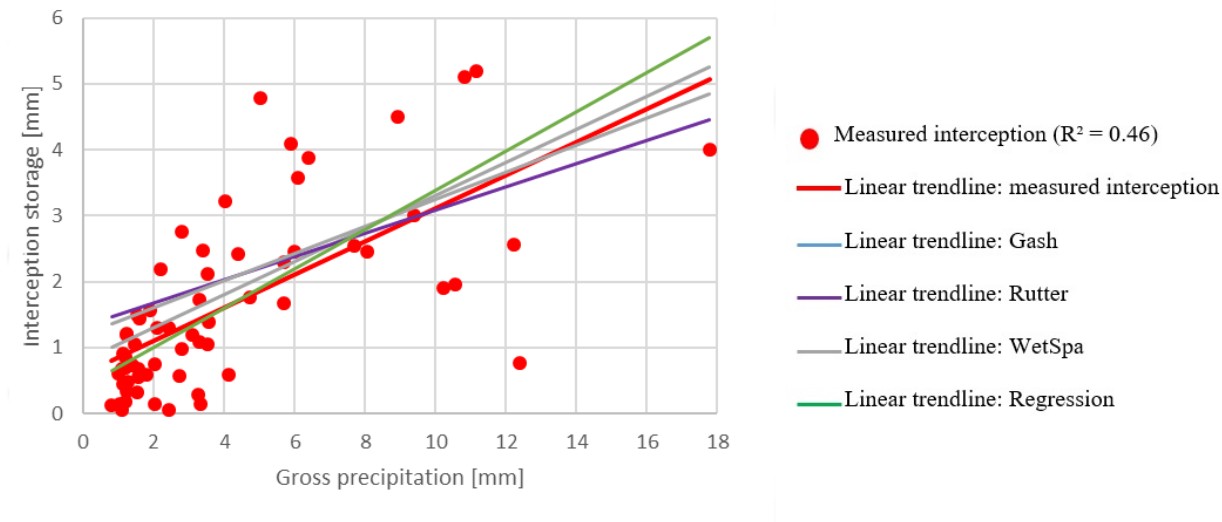

**Figure 12: Interception storage vs gross precipitation.**



**Table 6: Correlation between measured and simulated interception storage (R²/RMSE).**

| R² [%]/ $E_{RMS}$ [mm] | Gash | Rutter | WetSpa | Regression |
|---|---|---|---|---|
| All events (64 events) | 60/ 0.87 | 50/ 1.10 | 59/ 1.02 | **66/ 0.78** |
| Norway maple (39 events) | 62/ 0.78 | 49/ 1.10 | 52/ 1.06 | **81/ 0.52** |
| Small leaved lime (25 events) | 59/ 0.99 | 53/ 1.10 | **65/ 0.96** | 53/ 1.08 |
| Loosing leaves period (21 events) | **58/ 0.65** | 53/ 0.84 | 55/ 0.71 | 41/ 0.83 |
| Gaining leaves period (43 events) | 59/ 0.96 | 48/ 1.20 | 63/ 1.15 | **72/ 0.76** |
| Big events (Pg > 10mm) (7 events) | 30/ 1.44 | 44/ 1.28 | **46/ 1.25** | 5/ 1.79 |
| Small events (Pg < 10mm) (57 events) | 60/ 0.78 | 41/ 1.02 | 53/ 0.93 | **80/ 0.55** |





**Figure 13: Measured vs simulated interception storage of the Gash- (a), Rutter- (b) and WetSpa (c) model and the regression (d). (e) superimposes all four methods.**



### 3.7 The land cover change scenario

### 3.7.1 Scenario 1

5  In scenario 1 we assume that all the surfaces around the new student houses of the VUB Campus are sealed, resulting in an increase of the sealed surface by 54 % and a reduction of the tree area by 13 %. The simulation shows that this leads to an increase of the yearly net rainfall by 19 % (136 mm for 2015). The increased amount of rainfall reaching the ground surface is, in our simulations, distributed to depression storage and surface runoff (Fig. 14). The important increase in surface runoff volume (49 %, 352 mm) is due to the reduction of trees impeding interception and the sealing of pervious soil obstructing infiltration. Further the increase in depression storage might be higher than expected as we keep the original elevation map

10  where due to the more natural landscape depressions are more common. If we consider the runoff flux an increase of 147 % ( + 0.128 m³/s) is reached in scenario 1 (Fig. 15). Due to the small catchment area no redistribution in time of the peak runoff can be illustrated.

### 3.7.2 Scenario 2

In scenario 2 we assume that all the surfaces around the new student houses of the VUB Campus are replanted with trees (Norway maple). This leads to a tree cover of 35 % and an impervious cover of 61 %. This scenario shows that the high tree coverage restores interception storage to 18.4 % (130.31 mm) versus 21.5 % (152.53mm) before construction works started. The 24 % increase of sealed surfaces compared to the original situation results in an increase of 21 % of surface runoff, mostly

20  related to the obstruction of infiltration. The hydrograph in Fig. 15 of Scenario 2 shows an 64 % increase (+0.05 m³/s) of peak flow for the June event in 2015. The rainfall stored by tree interception is thus able to limit the increase in peak discharge to less than half of scenario 1.

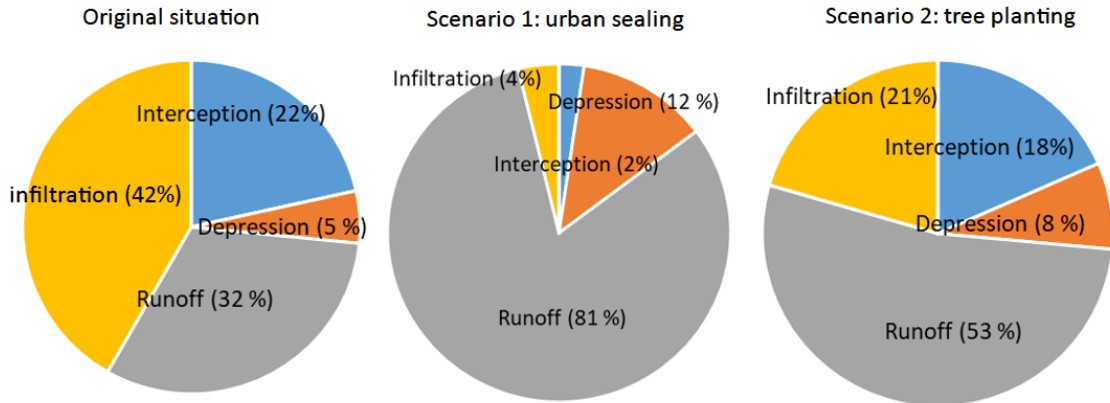

**Figure 14: Yearly (2015) surface water balance [% rainfall] for the different scenario's.**



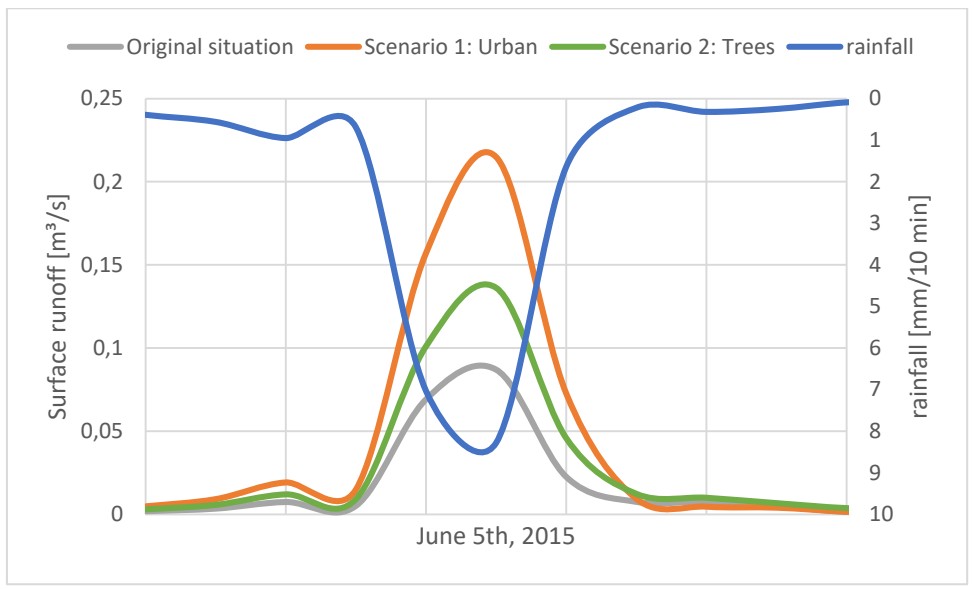

**Figure 15: hydrographs for 2015 event (return period 2 years Uccle).**

## 4. Discussion

### 4.1 The tree interception storage

Average interception storages during the measurement periods for both trees are, with 38.70 % for the Norway maple and 37.60 % for the small leaved lime, very similar. In general, the interception storage we measured is relatively high in comparison with other studies (Gómez et al., 2001; Staelens, 2010; Pereira et al., 2009; Xiao et al., 2000b; Xiao & McPherson, 2011). However, caution should be taken to directly compare interception storage as measurement conditions are different.

Our measurement periods only covered full leaf area- and transition periods. Defoliated trees were not monitored and, if included, these would lower the average amount of rain intercepted. There are however meaningful comparisons to be made with studies in Mediterranean climates. For example an evergreen *Gingko biloba* in the study of Xiao & McPherson (2011) and an evergreen *Quercus ilex* in the study of (Pereira et al., 2009) were found to intercept 25 % and 23 % of the gross precipitation respectively. Even though no defoliation occurred and leaf coverage was high, these interception storage values

are still far below the values measured in this study. In another study, Xiao et al. (2000a) derived interception storage values of a *Pyrus calleryana* 'Bradford' and a *Quercus suber* with constant LAI in a Mediterranean climate and found they intercepted 15 % and 27 % of the gross precipitation. The higher interception storage values we find indicate the suitability for tree interception in our temperate climate in comparison to the Mediterranean climate. Measured rain intensities in our study (Fig. 8) are low in comparison with those mentioned in studies performed in the Mediterranean area, where average intensities larger

than 2 mm/h are common (Pereira et al., 2009). Xiao et al. (2000a) found that when small rain events of low duration follow up on each other with a high frequency, the amount of intercepted water increases due to the consecutive wetting and drying of the crown surface.

We hence conclude that our initial hypothesis was correct and that trees in temperate climates are able to intercept a higher percentage of rainfall due to their more favorable rain event's characteristics (Cfr. Section 3.1).

### 4.2 The model comparisons





The Gash, Rutter and WetSpa models show similar performances. The regression equation performs best overall. However the difference in performance between the Norway maple and the small leaved lime indicates a strong tree dependency. Further, the difference between the seasons and the low performance for bigger rainfall events ($P_g > 10$ mm) also indicate that the regression method is not robust and therefore unsuitable to be used on a wider scale.

The model outputs are strongly depending on the interception storage capacity. Two methods to determine this variable have been investigated: the Leyton method (Leyton et al., 1967) and an LAI based equation (Gómez et al., 2001). The equation of Gómez is chosen for 3 reasons: (1) the equation has no subjective interpretation, (2) the equation is based on LAI which can be easily measured or retrieved from optical imagery and (3) because seasonal changes in interception storage can be taken into account. The latter is crucial for deciduous trees in temperate climates.

Further, we found that the simulations overestimate the interception measurements for small rainfall events ($P_g < 10$ mm) and small interception events (< interception storage capacity). This overestimation is occurring as some factors such as wind speed and orientation, rain intensity and inclination angle, leaf zenith angle and other meteorological and tree architectural parameters are not taken into account (Xiao et al., 2000b). For bigger interception events (> interception storage capacity) we simulate lower interception storage than we measure. We assume that the emptying of the storage by evaporation is higher than we
simulate with our models. Again a limitation of accounting for detailed meteorological and tree architectural parameters in our simulations might be the cause. Even though big interception events are most important for the retention of rainfall water on city trees, we still believe that this limitation is acceptable to limit the need of physical parameters in our simulations, as measuring all this parameters needs a lot of resources. The high similarity in performance of the specialized interception models (Gash and Rutter), the regression and the water balance model (WetSpa) let us conclude that WetSpa can be used for
studies on interception storage. To gain deeper understanding at a single tree level, more specialized interception models such as the method of Xiao et al. (2000b) might be more recommendable. However, as we could not measure all the parameters needed for this method, we were not able to evaluate its performance for our trees, which indicates the operational limitations of this model.

### 4.3 The land cover change scenario's

Our scenario analysis of a small construction site on the VUB campus (< 0.02 km²) shows that by replacing this small green area with new buildings the water balance is highly disturbed (scenario 1). As net rainfall increases and the infiltration capacity is limited, due to the sealing of surfaces, a high amount of surface runoff is produced. For a rainfall event with a return period of 2 years the increase in surface runoff is of 147 % of the initial runoff flux. Such an increase of surface runoff on a small
scale accumulates over an urban catchment through the connectivity of imperviousness (Verbeiren et al., 2013) and challenges the capacities of installed drainage systems and retention reservoirs. To avoid such scenarios it is important to mitigate the effects of new constructions. Therefore we considered a second scenario where surroundings of buildings are kept permeable and are covered with trees. This scenario almost restores the interception capacity of the area. The retention of net rainfall due to tree interception limits the effect the increased sealed surfaces (+24 %) have on the surface runoff. The increase of the peak
discharge for the event with return period 2 years is limited by tree interception to less than half of scenario 1. However, even this unrealistic scenario of planting trees all around the buildings doesn't allow to re-establish the water balance and additional compensation measures would be needed (green roofs, compensation off-site, etc.).





## 5.Conclusions

To evaluate the importance of city trees for reducing the net rainfall in a temperate climate we evaluated (1) in-situ interception experiments and (2) different interception simulation tools on two solitary trees. Further we performed a scenario analysis to evaluate the effect land cover changes on the water balance of a small scale pilot case.

Our main conclusions are:

(1) With 38 % of the gross precipitation, the intercepted rainfall was higher than observed for solitary trees in Mediterranean
climates. Trees in temperate climates are believed to be more capable of intercepting rain water due to the more favorable rain event characteristics such as a longer event duration and a lower rain intensity.

(2) The water balance model (WetSpa) performed similarly to the specialized interception models of Gash and Rutter. The Gash model performs better than the Rutter model on the individual tree level in temperate climates. The performance of WetSpa is in between both interception models and best for bigger rainfall events ($P_g > 10$ mm).

(3) The regression equation we developed for the interception storage in temperate climates was found to be tree- and season-specific. We therefore don't recommend its use  for a wider application as its performance is limited by the amount of meteorological data and by the tree architectural characteristics.

(4) The scenario analysis shows that even small scale construction sites can have an impact on the water management in cities. In our scenario a green space is replaced by new student houses on the VUB campus in Brussels, Belgium. This change increases
net rainfall by 20 % and surface runoff by 49 %. A more sustainable scenario where all areas around buildings are covered by trees re-establishes the interception storage and thus limits net rainfall. However, due to less pervious surfaces, infiltration is still limited and surface runoff increased. This scenario analysis gives an indication of how city trees retain rainfall water and limit surface runoff production.

**Appendix A: Model equations:**

$$I = S + E = P_g - (T_f + D + S_f)$$                          equation 1 (Xiao, 1998)

$I$: interception storage [mm]
$S$ : crown surface storage capacity = interception storage capacity [mm]
$E$ : evaporation [mm]
$P_g$ : gross rainfall [mm]
$T_f$: free trough-fall [mm]
$D$ : drip-off [mm]
$S_f$: stemflow [mm]

$$T_f = P \times P_g$$                                 equation 2 (Xiao, 1998)

$P$  = gap fraction [-]




$$\begin{cases} \mathbf{D} = \mathbf{0} & \text{for I < S} \\ \mathbf{D} = \mathbf{I} - \mathbf{S} & \text{for I} \geq \text{S} \end{cases}$$ equation 3 (Valente et al., 1995)

**Gash**

$$\begin{cases} \mathbf{I} = (1\text{-}p) \times P_g & \text{for P< P'} \\ \mathbf{I} = (1\text{-}p) \times \text{P'} + \frac{Ea}{R} \times (P_g\text{-P'}) & \text{for P> P'} \end{cases}$$ equation 5 (Gash, 1979)

$Ea$ : mean evaporation / canopy cover [-]
**R**: mean rainfall/ saturated canopy cover [-]
10 **P'**: precipitation reaching canopy saturation [mm]

$$\mathbf{P'} = -\frac{R}{Ea} \times \mathbf{S} \times ln(1 - \frac{Ea}{R\times(1-p)})$$ equation 6 (Gash, 1979)

**Rutter**

$$\mathbf{P}_{soil} = T_f + \mathbf{D} + S_f$$

and

$$\begin{cases} \mathbf{P}_{soil} = P_g - \mathbf{E} - (\mathbf{S}\text{-IS}) & \text{for P - E > C – IS} \\ \mathbf{P}_{soil} = \mathbf{0} & \text{for P - E <= C - IS} \end{cases}$$ equation 7 (Vegas et al., 2012)

**P**soil **:** precipitation reaching the ground (net precipitation) [mm]
**IS:** interception storage at timestep before actual rainfall [mm]

$$\mathbf{E} = \frac{IS}{S} \times \mathbf{PET}$$ equation 8 (Vegas et al., 2012)

**PET**: Potential evapotranspiration estimated with the Penman-Monteith equation (Monteith, 1965).

$$\mathbf{I} = \mathbf{E} + (\mathbf{S}\text{- IS})$$ equation 9 (Vegas et al., 2012)

**WetSpa**

$$\begin{cases} \mathbf{I} = \mathbf{S} - \mathbf{IS} & \text{for P > C – IS} \\ \mathbf{I} = P_g & \text{for P <= C- IS} \end{cases}$$ equation 10 (WetSpa manual, 2009)

and

$$\mathbf{IS} (t) = \mathbf{IS} (t\text{-}1) + \mathbf{I} (t) – \mathbf{E} (t)$$ equation 11 (WetSpa manual, 2009)

and

$$\begin{cases} \mathbf{E}(t) = \mathbf{IS}(t\text{-}1) & \text{for PET > IS(t-1)} \\ \mathbf{E} (t) = \mathbf{PET} & \text{for PET < IS(t-1)} \end{cases}$$ equation 12 (WetSpa manual, 2009)

40

**Appendix B: equations Figure 9.**





Figure 9a: Uncertainty equation Norway maple: $\qquad y = 0.3884 \times x^{-1}$

Uncertainty equation small leaved lime: $\qquad y = 0.3468 \times x^{-1}$

Figure 9b: Sensivity equation Norway Maple: $\qquad y = 3.4658 \times x^{-1}$

Sensivity equation small leaved lime: $\qquad y = 3.097 \ \times x^{-1}$

**Appendix C**: Clustering variables regression equation

| Cluster 1 | Cluster 2 | Cluster 3 | Cluster 4 | Cluster 5 |
|---|---|---|---|---|
| $\boldsymbol{P_g}$ | **LAI** | PET | **Duration** | Max. wind speed |
| Min. wind speed | Min. T | Min. Humidity | $P_w$ | Av. wind speed |
| Rel. wind speed | Max. T | Max Humidiy | Intensity | |
| W | Av. T | Av. Humidity | IERI | |

$P_g$ = Gross precipitation; W = Relative windspeed/$P_g$ ; PET = Potential evapotranspiration; $P_w$ = $P_g$ /intensity; Relative windspeed = av. windspeed/max. windspeed

10   Cluster 3 and 5 were eliminated from the regression. The variables chosen for the regression in Cluster 1,2 and 4 are highlighted in bold.

**Appendix D**: Individual rain events

| Event nr. | Date | Tree | Event start | Event duration (h:mm:ss) | Pg (mm) | Intensity (mm) | IERI | LAI | Wind speed (m/s) | I (mm) | TF (mm) | ST (mm) | I (%) | TF (%) | ST (%) |
|---|---|---|---|---|---|---|---|---|---|---|---|---|---|---|---|
| 1 | 14/08/2015 | Norway maple | 15:47:14 | 0:14:01 | 1,1 | 4,71 | 0,00 | 3,4 | 1,50 | 0,65 | 0,45 | | 0,59 | 0,41 | |
| 2 | 17/08/2015 | Norway maple | 20:17:50 | 8:09:21 | 7,7 | 0,94 | 0,45 | 3,3 | 0,52 | 2,46 | 5,24 | | 0,32 | 0,68 | |
| 3 | 23/08/2015 | Norway maple | 16:32:22 | 1:50:56 | 1,2 | 0,65 | 0,43 | 3,1 | 0,70 | 0,68 | 0,52 | | 0,57 | 0,43 | |
| 4 | 24/08/2015 | Norway maple | 11:30:26 | 15:48:11 | 6,4 | 0,4 | 0,88 | 3 | 2,09 | 3,73 | 2,67 | | 0,58 | 0,42 | |
| 5 | 26/08/2015 | Norway maple | 20:46:25 | 6:03:25 | 2,1 | 0,35 | 0,86 | 2,9 | 1,29 | 1,18 | 0,92 | | 0,56 | 0,44 | |
| 6 | 31/08/2015 | Norway maple | 17:56:53 | 0:14:25 | 1 | 4,16 | 0,00 | 2,8 | 1,15 | 0,58 | 0,42 | | 0,58 | 0,42 | |
| 7 | 1/09/2015 | Norway maple | 7:05:54 | 1:03:38 | 1,2 | 1,13 | 0,54 | 2,7 | 1,10 | 0,83 | 0,37 | | 0,69 | 0,31 | |
| 8 | 12/09/2015 | Norway maple | 14:20:24 | 9:31:25 | 6,1 | 0,64 | 0,63 | 2,3 | 0,68 | 3,46 | 2,64 | | 0,57 | 0,43 | |
| 9 | 16/09/2015 | Norway maple | 4:00:16 | 2:49:09 | 3,3 | 1,17 | 0,29 | 2,2 | 1,30 | 1,05 | 2,25 | | 0,32 | 0,68 | |
| 10 | 18/09/2015 | Norway maple | 18:41:02 | 2:45:05 | 1,9 | 0,69 | 0,79 | 2,1 | 0,88 | 1,51 | 0,39 | | 0,8 | 0,2 | |
| 11 | 21/09/2015 | Norway maple | 21:02:11 | 4:57:19 | 1,6 | 0,32 | 0,77 | 2 | 1,30 | 1,39 | 0,21 | | 0,87 | 0,13 | |
| 12 | 7/10/2015 | Norway maple | 12:08:40 | 7:07:42 | 1,8 | 0,25 | 0,82 | 1,4 | 1,56 | 0,56 | 1,24 | | 0,31 | 0,69 | |
| 13 | 14/10/2015 | Norway maple | 12:01:27 | 8:26:18 | 5,7 | 0,68 | 0,54 | 1 | 0,96 | 1,61 | 4,09 | 0,07 | 0,28 | 0,72 | 0,01 |
| 14 | 24/03/2016 | Norway maple | 22:35:00 | 12:55:00 | 9,4 | 0,73 | 0,06 | 0,6 | 1,65 | 3 | 6,38 | 0,05 | 0,32 | 0,68 | 0,01 |
| 15 | 29/03/2016 | Norway maple | 12:45:00 | 2:20:00 | 2,03 | 0,87 | 0,64 | 0,6 | 1,35 | 0,14 | 1,87 | | 0,07 | 0,92 | |
| 16 | 30/03/2016 | Norway maple | 20:45:00 | 8:20:00 | 4,12 | 0,49 | 0,25 | 0,6 | 0,58 | 0,59 | 3,53 | | 0,14 | 0,86 | |
| 17 | 5/04/2016 | Norway maple | 2:00:00 | 3:05:00 | 1,09 | 0,35 | 0,22 | 0,7 | 0,84 | 0,05 | 1,04 | | 0,05 | 0,95 | |
| 18 | 9/04/2016 | Norway maple | 22:00:00 | 4:25:00 | 1,52 | 0,34 | 0,30 | 0,7 | 1,01 | 0,32 | 1,2 | | 0,21 | 0,79 | |
| 19 | 13/04/2016 | Norway maple | 15:25:00 | 1:00:00 | 3,31 | 3,31 | 0,00 | 0,8 | 1,13 | 0,14 | 3,17 | | 0,04 | 0,96 | |
| 20 | 23/04/2016 | Norway maple | 1:40:00 | 3:35:00 | 1,12 | 0,31 | 0,40 | 1,1 | 2,27 | 0,45 | 0,67 | | 0,4 | 0,6 | |
| 21 | 28/04/2016 | Norway maple | 10:25:00 | 2:20:00 | 1,02 | 0,44 | 0,82 | 1,9 | 1,34 | 0,14 | 0,88 | | 0,14 | 0,86 | |
| 22 | 29/04/2016 | Norway maple | 9:05:00 | 6:50:00 | 2,8 | 0,41 | 0,37 | 2 | 1,56 | 0,98 | 1,82 | | 0,35 | 0,65 | |
| 23 | 30/04/2016 | Norway maple | 22:15:00 | 3:35:00 | 1,57 | 0,44 | 0,16 | 2 | 0,64 | 0,67 | 0,9 | | 0,43 | 0,57 | |
| 24 | 9/05/2016 | Norway maple | 23:00:00 | 18:15:00 | 4,02 | 0,22 | 0,74 | 3,2 | 0,91 | 3,21 | 0,81 | | 0,8 | 0,2 | |
| 25 | 22/05/2016 | Norway maple | 3:40:00 | 9:55:00 | 3,4 | 0,34 | 0,55 | 3,4 | 0,95 | 2,46 | 0,94 | | 0,72 | 0,28 | |
| 26 | 27/05/2016 | Norway maple | 19:15:00 | 2:10:00 | 5,69 | 2,63 | 0,08 | 3,5 | 0,78 | 2,29 | 3,4 | | 0,4 | 0,6 | |
| 27 | 1/06/2016 | Norway maple | 11:55:00 | 7:05:00 | 3,52 | 0,5 | 0,74 | 3,5 | 0,88 | 2,12 | 1,4 | | 0,6 | 0,4 | |
| 28 | 2/06/2016 | Norway maple | 8:15:00 | 4:00:00 | 1,46 | 0,37 | 0,46 | 3,5 | 1,34 | 1,04 | 0,42 | | 0,71 | 0,29 | |
| 29 | 3/06/2016 | Norway maple | 20:05:00 | 5:45:00 | 3,52 | 0,61 | 0,33 | 3,5 | 0,89 | 1,05 | 2,47 | | 0,3 | 0,7 | |
| 30 | 13/06/2016 | Norway maple | 18:25:00 | 0:10:00 | 1,23 | 7,38 | 0,00 | 3,6 | 1,09 | 1,2 | 0,03 | | 0,98 | 0,03 | |
| 31 | 15/06/2016 | Norway maple | 10:25:00 | 3:30:00 | 10,21 | 2,92 | 0,43 | 3,6 | 0,79 | 1,9 | 8,31 | | 0,19 | 0,81 | |
| 32 | 17/06/2016 | Norway maple | 9:00:00 | 7:05:00 | 2,71 | 0,38 | 0,73 | 3,6 | 0,94 | 0,56 | 2,15 | | 0,21 | 0,79 | |
| 33 | 18/06/2016 | Norway maple | 3:40:00 | 12:15:00 | 10,81 | 0,88 | 0,79 | 3,6 | 0,83 | 5,11 | 5,67 | 0,06 | 0,47 | 0,52 | 0,01 |
| 34 | 21/06/2016 | Norway maple | 7:55:00 | 10:15:00 | 5,99 | 0,58 | 0,61 | 3,6 | 0,90 | 2,45 | 3,54 | | 0,41 | 0,59 | |
| 35 | 23/06/2016 | Norway maple | 19:20:00 | 5:30:00 | 12,24 | 2,23 | 0,36 | 3,6 | 0,74 | 2,56 | 9,68 | | 0,21 | 0,79 | |
| 36 | 2/07/2016 | Norway maple | 9:45:00 | 3:35:00 | 3,08 | 0,86 | 0,77 | 3,6 | 1,31 | 1,19 | 1,89 | | 0,39 | 0,61 | |
| 37 | 3/07/2016 | Norway maple | 10:45:00 | 1:00:00 | 3,56 | 3,56 | 0,00 | 3,6 | 1,00 | 1,38 | 2,18 | | 0,39 | 0,61 | |
| 38 | 28/07/2016 | Norway maple | 6:15:00 | 0:35:00 | 1,18 | 2,02 | 0,00 | 3,6 | 0,89 | 0,17 | 1,01 | 0,01 | 0,14 | 0,86 | 0,01 |
| 39 | 29/07/2016 | Norway maple | 22:40:00 | 2:15:00 | 2,01 | 0,89 | 0,22 | 3,6 | 1,41 | 0,75 | 1,26 | | 0,37 | 0,63 | |
| 40 | 29/09/2016 | Small leaved lime | 20:30:00 | 1:45:00 | 1,56 | 0,89 | 0,29 | 2,7 | 6,61 | 0,55 | 1,01 | | 0,35 | 0,65 | |
| 41 | 1/10/2016 | Small leaved lime | 19:30:00 | 1:30:00 | 3,29 | 2,19 | 0,50 | 2,5 | 5,64 | 1,73 | 1,56 | | 0,53 | 0,47 | |
| 42 | 12/10/2016 | Small leaved lime | 6:45:00 | 4:30:00 | 1,27 | 0,28 | 0,56 | 1,9 | 4,41 | 0,48 | 0,79 | | 0,38 | 0,62 | |
| 43 | 14/10/2016 | Small leaved lime | 20:45:00 | 11:45:00 | 12,38 | 1,05 | 0,00 | 1,7 | 5,11 | 0,76 | 11,6 | | 0,06 | 0,94 | |
| 44 | 2/11/2016 | Small leaved lime | 20:00:00 | 4:45:00 | 1,39 | 0,29 | 0,58 | 0,5 | 4,44 | 0,73 | 0,66 | | 0,52 | 0,48 | |
| 45 | 5/11/2016 | Small leaved lime | 22:45:00 | 5:00:00 | 1,3 | 0,26 | 0,85 | 0,5 | 6,20 | 0,75 | 0,55 | | 0,57 | 0,43 | |
| 46 | 7/11/2016 | Small leaved lime | 12:30:00 | 5:15:00 | 2,43 | 0,46 | 0,38 | 0,5 | 4,27 | 0,06 | 2,37 | | 0,02 | 0,98 | |
| 47 | 12/11/2016 | Small leaved lime | 20:45:00 | 11:30:00 | 3,27 | 0,28 | 0,30 | 0,5 | 3,13 | 0,29 | 2,98 | | 0,09 | 0,91 | |
| 48 | 15/04/2017 | Small leaved lime | 6:30:00 | 6:15:00 | 2,42 | 0,39 | 0,32 | 1,2 | 6,16 | 1,29 | 1,13 | | 0,53 | 0,47 | |
| 49 | 16/04/2017 | Small leaved lime | 21:15:00 | 13:00:00 | 8,94 | 0,69 | 0,54 | 1,3 | 5,79 | 4,5 | 4,4 | 0,03 | 0,5 | 0,49 | 0 |
| 50 | 25/04/2017 | Small leaved lime | 1:00:00 | 3:00:00 | 1,38 | 0,46 | 0,08 | 2,1 | 5,66 | 0,22 | 1,16 | | 0,16 | 0,84 | |
| 51 | 2/05/2017 | Small leaved lime | 12:15:00 | 20:30:00 | 5,9 | 0,29 | 0,44 | 3 | 5,12 | 4,08 | 1,82 | | 0,69 | 0,31 | |
| 52 | 18/05/2017 | Small leaved lime | 3:30:00 | 7:15:00 | 4,4 | 0,61 | 0,00 | 3,9 | 3,81 | 2,43 | 1,97 | | 0,55 | 0,45 | |
| 53 | 19/05/2017 | Small leaved lime | 0:15:00 | 1:30:00 | 1,22 | 0,81 | 0,17 | 4 | 5,84 | 0,34 | 0,88 | | 0,28 | 0,72 | |
| 54 | 9/06/2017 | Small leaved lime | 6:15:00 | 4:15:00 | 11,16 | 2,63 | 0,00 | 4,3 | 4,35 | 5,18 | 5,98 | | 0,46 | 0,54 | |
| 55 | 25/06/2017 | Small leaved lime | 2:00:00 | 4:00:00 | 1,52 | 0,38 | 0,44 | 4,5 | 6,16 | 1,5 | 0,02 | | 0,99 | 0,01 | |
| 56 | 27/06/2017 | Small leaved lime | 20:15:00 | 4:00:00 | 8,05 | 2,01 | 0,25 | 4,5 | 4,41 | 2,45 | 5,6 | | 0,3 | 0,7 | |
| 57 | 1/07/2017 | Small leaved lime | 3:30:00 | 14:30:00 | 17,78 | 1,23 | 0,12 | 4,6 | 5,99 | 3,99 | 13,8 | 0,08 | 0,22 | 0,77 | 0 |
| 58 | 6/07/2017 | Small leaved lime | 14:00:00 | 1:00:00 | 1,23 | 1,23 | 0,00 | 4,7 | 4,46 | 1,21 | 0,02 | | 0,98 | 0,02 | |
| 59 | 17/07/2017 | Small leaved lime | 9:00:00 | 3:00:00 | 2,2 | 0,73 | 0,17 | 4,8 | 4,92 | 2,18 | 0,02 | | 0,99 | 0,01 | |
| 60 | 22/07/2017 | Small leaved lime | 19:00:00 | 7:15:00 | 2,79 | 0,38 | 0,69 | 4,8 | 5,78 | 0 | 0 | | | | |
| 61 | 24/07/2017 | Small leaved lime | 8:30:00 | 12:45:00 | 5,01 | 0,39 | 0,71 | 4,8 | 4,10 | 4,78 | 0,23 | | 0,96 | 0,05 | |
| 62 | 8/08/2017 | Small leaved lime | 23:15:00 | 0:45:00 | 1,13 | 1,51 | 0,00 | 4,8 | 5,89 | 0,91 | 0,22 | | 0,81 | 0,19 | |
| 63 | 15/08/2017 | Small leaved lime | 8:45:00 | 2:30:00 | 4,72 | 1,89 | 0,40 | 4,8 | 7,21 | 1,76 | 2,96 | | 0,37 | 0,63 | |
| 64 | 18/08/2017 | Small leaved lime | 3:30:00 | 11:45:00 | 10,57 | 0,9 | 0,62 | 4,8 | 5,90 | 1,95 | 8,62 | | 0,18 | 0,82 | |



**Author Contributions:** V.S., C.W., B.S. and B.V. conceived and designed the ground-truthing experiments and the simulation strategy. V.S. and C.W. set-up the V-catchment and performed the ground-truthing measurements. V.S. took the lead in the analysis of the measurement results and C.W. fulfilled the simulations and scenario analysis. V.S., C.W., B.S. and B.V analyzed the results, and prepared the structure of the manuscript. V.S. and C.W. wrote the initial draft of the paper. W.B.,
M.H., B.S. and B.V supervised the research and contributed to improving the manuscript prior to submission.

**Conflicts of Interest:** The authors declare no conflict of interest.

**Acknowledgments:** This research is co-funded within the framework of the UrbanEARS project SR/00/307 from the Belgian
Federal Science Policy Office, Support to the Exploitation and Research in Earth Observation III (BELSPO STEREOIII), and the Belgian airborne calibration and validation sites for urban and forest (BELAIR-SONIA) project SR/03/333. This research is also funded by Fonds Wetenschappelijk Onderzoek Vlaanderen (FWO) [FWO-SB, no. 121124]. The meteorological dataset is provided by the Royal Meteorological Institute (Uccle) and Flowbru (depot communal). We also want to thank Msc Sarah Mommers and Msc Anne-Sophie Mulier together with the technical assistants Christophe Coeck, Eric Van Beek and Remi
Chevalier for their contribution to the measurement campaigns

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
