# Peer review of "The importance of city trees for reducing net rainfall: comparing measurements and simulations"

_Hydrology and Earth System Sciences, 2018_

## Referee Comment (RC1) · Anonymous Referee #1 · 17 Oct 2018

Review comments The importance of city trees for reducing net rainfall: comparing measurement and simulations By V. Smets, C. Wirion, W. Bauwes, M. Hermy, B. Somers, B. Verbeiren

The hydrological role of urban vegetation is an issue of growing importance, in relation with urban sustainable development, and more recently in relation with urban climatology. This manuscript which deals with urban trees addresses both societal concerns, which are also scientific concerns, urban vegetation being a widely open research subject. The authors study interception by urban trees. The manuscript combines field experiment and modelling complemented by a case study. The field experiment

seems to me original, the model comparison concerns existing models, applied to an urban context, and I consider that this manuscript is of interest to HESS. Nevertheless, I would suggest a significant revision before its publication :

- The manuscript is, in my opinion, too long and contains details which are not very useful and which don't help the reading. I would recommend a shorter manuscript, written in a more synthetic and efficient way and focusing on the key points of the contribution, and may more rigorous (notation and denomination of variables, model presentation) - The manuscript organisation could be simplified. Each section is divided into many paragraphs (up to 9 or 10) which don't help the understanding. - The field experiment concerns individual trees. What about the tested models? Do they apply to individual trees or tree covers? Consequences concerning the model evaluation?

The following comments should help the authors to revise the manuscript.

p 1 – line 33-36: if the problem of heavy events should not be forgotten, the growing interest for vegetation in urban areas seems more related to the promotion of sustainable urban development, and more recently to urban climatology. p 2 – paragraph 1: I have some agreement to that. The authors may indicate that the role of vegetation in urban areas remains a very open question, largely unstudied so far. p 3 – paragraph 1.3: in urban areas, many trees are tree lines, along streets. In that case, the soil around these trees is sealed (impervious). Is the situation studied by the authors really representative of urban trees? p 5-11: The experiment device is interesting and original. Nevertheless, it raises issues. How representative is this experimental setting of urban trees that are subject to very different contexts ?. I would suggest that the authors write in a more synthetic way. p 10 – paragraph 2.6: what difference between data processing and modelling (next section ? It is not clear to me. p 12 – paragraph 2: If modelling and model comparison is a central part of the manuscript, I would suggest to present the models as part of the manuscript and not in Appendix. p 12 – paragraph 2.8: Do these models apply to individual trees, as the field experiment, or a tree plant cover? I suppose that there is a very important difference between the two. It is a

significant point concerning the model relevance. p 13: the scenario analysis appears as important as the modelling and model comparison. Is it really the case, or is it just an illustration example, in which case, I would suggest to reduce its length. p 15 – lines 5 to 14: I would suggest to move this lines to Section 2. p 17 – figure 8: I am not convinced that this figure is very useful here, it could eventually be moved in Appendix. p 19 – paragraph 3.3: is a full page needed to see the weak sensitivity? A few lines would be enough. p 20: line 1-4: small events of 1 mm are not very important from a hydrological point of view p 20: figure 10: the presence of the inflexion point on the regression lines is not obvious. Is it justified by physical reasons? p 22: Equation 6. The authors introduce I as interception (in mm). The legend of Table 5 mentions the Interception storage capacity (S in mm), as Equation 5 (p 11) and in Appendix A (Eq. 1) I is the interception storage, and S the crown surface storage capacity, and IS appears in following equations. In equation 6, D is duration and in A-1, it is drip-off . . ... A very careful checking of notations is required. What is the difference between interception, interception storage and storage capacity. p 22: Equation 6 – I am sceptical concerning the regression model (eq. 6) which provides better results, but which introduces 9 numbers, that is 9 parameters, which means that this model is strictly limited to the analysed data set, without any generalization. Such a model is of a very limited interest. p 23: Figure 12 – Interception storage – A new denomination? p 25: I would suggest that Figure 13 and Table 6 to be the central points of the model comparison. p 26-27 paragraph 3.7: I wonder about the usefulness of this paragraph "land cover change scenario" in this manuscript. In the light of previous results, it is not surprising that an increase of sealed surfaces generates a runoff increase, and that tree planting generates an increase of water storage. p 27: It seems to me that interesting discussion subject could have been the following: i) validity of this model assessment given the possible differences between individual trees and tree plant cover; ii) application of the models to a more usual urban context: available data, types of trees . . ... p 32: Table to remove, it is unreadable (the data set can be provided on request).

---

## Referee Comment (RC2) · Anonymous Referee #2 · 7 Nov 2018

This manuscript describes some useful work to estimate canopy interception from urban trees. The experimental data appear to generally follow expectations, and conclusions are apparently well supported.

The manuscript is, however, unusually long, and contains extended reviews of basic and standard canopy interception concepts that are not necessary and make the original contributions of the work difficult to extract. For example, Figures 1 and 5 are unnecessary: these have been standard for 50 years, and also the same analysis as Figure 5 was done for this research and presented as Figure 10. Sections 1.2, 1.3, and 2.6 could all be reduced substantially.

[Figure]

The choice of and behavior of WetSpa for interception modeling is difficult to evaluate. The WetSpa equations in Appendix A contain symbols that are not defined and the cited source for these is a white paper not contained in the References. As far as I can determine the simple assumptions it makes have never been tested in the refereed literature, despite the citations to support use of the WetSpa interception formulation (e.g., P5L15) that do establish that it has been used. There appears to be no scientific reason to include this model in a comparison, and it seems likely that it was chosen because of its familiarity to the authors.

The simplicity of the WetSpa formulation as an ultra-simple bucket model with, e.g., no provision for drip when storage is less than capacity or reduced evaporation rates from partially wetted canopy, is an interesting test against the more sophisticated Gash and Rutter models. That it appears to give empirically similar or superior results is a very useful finding for canopy interception science. However, the discussion makes essentially no attempt to explore the ramifications of this finding, either from a utilitarian (e.g., "should we be using simpler models") or physical (e.g., "what does it mean that only the coarsest components of the water budget need modeling?" Or "Is the superior performance of the super-simple model a fluke of this environment or should we be applying it more generally?") perspective. Re-stating and discussing the "WetSpa" formulation in terms of other, similar models in the older literature would help it to make the biggest contribution to the canopy interception literature.

I think the urban hydrology modeling is a separate topic that is best left for another paper. Removing it would help shorten the paper to a more manageable size and also allow the strengths of the interception data to be better emphasized. Perhaps it is only a matter of taste, but I think the style in which the urban hydrological modeling text is written suggests bias on the part of the authors about the importance of urban trees, when a dispassionate evaluation would be more effective.

The canopy interception conclusions mostly consist either of simple data or bland inferences based more on previous understanding of urban hydrology than on the results

of this work. Adding theoretical discussion and making theoretical inferences from the data would help readers understand the scope of the work beyond the immediate context of this experiment.

A few detailed comments: P12L15 Salvadore et al. 2015 published a general review of models of urban hydrology, and it is not clear how that review supports this work.

Figure 2 is not needed; its function is duplicated by Figure 3.

Section 3.1 should be Methods.

Sections 2.5.2 and 3.3 the sensitivity analysis is difficult to understand. It seems like the data in the two panels of Figure 9 probably came from the same assumptions, but I cannot follow the text P9L34-37

Figure 10 the inflection point is plotted at $\sim$2.5 but listed as 4.72.

Figure 13e is unnecessary. The same information is in a-d. The equations for the regressions should be presented.

Table 2 I disagree that regression does not account for evaporation during the event. It simply does so implicitly.

---

## Author Response (AR1)

**Response Reviewer 1**

**Authors:** Thank you for reviewing our paper. We believe that your comments have significantly strengthened the manuscript.

**Reviewer main comment 1:** The manuscript is, in my opinion, too long and contains details which are not very useful and which don't help the reading. I would recommend a shorter manuscript, written in a more synthetic and efficient way and focusing on the key points of the contribution, and may more rigorous (notation and denomination of variables, model presentation).

**Authors response:** We agree and shortened the paper from 10340 words to 8006 words (abstract + main text + conclusions). The whole manuscript went from 33 to 24 pages, including references and appendices. Many sections such as the materials and methods, and results section were shortened. The scenario analysis and regression were removed to focus on the key points of the manuscript.

**Change in manuscript:** See manuscript

**Reviewer main comment 2:** The manuscript organization could be simplified. Each section is divided into many paragraphs (up to 9 or 10) which don't help the understanding. - The field experiment concerns individual trees. What about the tested models? Do they apply to individual trees or tree covers? Consequences concerning the model evaluation ?

**Authors response:** We simplified the manuscript, eliminated an merged several paragraphs, removed unnecessary figures, etc. We clarified which models are used for forest cover interception and which models are used for individual trees (Table 4). We discussed the differences between these models more in detail. More elaborated answers are written in response to the line comments.

**Change in manuscript:** See manuscript and line comments.

**Line comments**

**Reviewer comment:** p 1 – line 33-36: if the problem of heavy events should not be forgotten, the growing interest for vegetation in urban areas seems more related to the promotion of sustainable urban development, and more recently to urban climatology.

**Authors response:** We agree that there are probably other drivers promoting the incorporation of vegetation in urban areas such as for example heat reduction and personal wellbeing.

In case of hydrological ecosystem services, vegetation only functions as a part of the solution and is most functional in the case of less intense and more spread out rainfall events. These events are the most common in our study (average intensity 1.3 mm/h for the Norway maple site and 0.9 mm/h for the small leaved lime site in comparison with an average > 2mm/h in Mediterranean climates (Pereira et al., 2009)).
In the case of heavy rainfall events the storage capacity of trees and smaller vegetation will most likely be exceeded, limiting the influence of the vegetation. However, even in the case of heavy

rainfall events, peak runoff will be delayed and spread out, resulting in less intense pressure on the drainage system (Szota et al., 2019).

**Change in manuscript:** We decided to not change the text. On p 2 – line 3 we start with 'part of the solution', emphasizing that incorporating urban green is not enough and complementary measures are also necessary. Furthermore, on p 1 – line 37-38, we say that an important challenge is an efficient water regulation policy, which includes more than just runoff reduction. We think urban green has an important part in such a policy.

**Reviewer comment:** p 2 – paragraph 1: I have some agreement to that. The authors may indicate that the role of vegetation in urban areas remains a very open question, largely unstudied so far.

**Authors response:** We don't think it is largely unstudied. Actually the influence of vegetation on the urban hydrological cycle has been a booming topic in recent years. For example when looking at google scholar: Inserting the keywords 'urban' 'vegetation' and 'hydrology' results in 17400 results since 2015.
We do think however that there are still some knowledge gaps to address. For example, the most commonly used models that estimate interception storage are stand-alone interception models (Gash, Rutter) that are not incorporated in a complete water balance model. That's why we compare these specialized models with the interception storage estimates of WetSpa, a model capable of modelling the whole water balance. Another knowledge gap is that most interception studies are done in Mediterranean climates, with different rainfall distribution patterns and vegetation. We want to know if the interception potential of trees in temperate climates is different than in Mediterranean climates.

**Change in manuscript:** We rewrote the last paragraph of section 1.2 to better emphasize the knowledge gaps we want to address and rewrote the research questions in section 1.6.

**Reviewer comment:** p 3 – paragraph 1.3: in urban areas, many trees are tree lines, along streets. In that case, the soil around these trees is sealed (impervious). Is the situation studied by the authors really representative of urban trees?

**Authors response:** in the case of the interception process, our experimental setup is representative, as it is a solitary tree and the ground surface does not influence the interception process. Off course due to the heterogeneous nature of urban environments, trees are found in widely varying settings and cannot all be reproduced in an experimental setup (proximity to buildings and other trees, varying wind directions, sun exposure, etc). We believe it is important to simulate interception storage for such a variety of urban trees and thus propose to use a LAI-based approach to calculate interception storage with a water balance model. The LAI based interception calculation accounts for the tree development/health in the urban environment and the distributed water balance model accounts for further urban constraints in the infiltration, evaporation and runoff calculation (imperviousness, connectivity of water flow, etc.). By placing our experimental setup on a site that is free from obstructions such as buildings and other trees, we believe we represent an ideal case

for urban trees that is adaptable to more specific conditions, providing that some assumptions are made.

**Change in manuscript:** added the text: **'**These two trees represent an urban solitary tree. As the urban environment is very heterogeneous, trees are found in widely varying settings such as in parcs, private gardens and on streets. Due to the limitations of an experimental setup (safety, space and logistics) we decide to choose urban solitary trees free from obstructions and with full sun- and wind exposure. The results of our experiment can thus not simply be translated to other solitary urban trees but must undergo some assumptions of the environmental conditions.' ' to section 2.2.

**Reviewer comment:** p 5-11: The experiment device is interesting and original. Nevertheless, it raises issues. How representative is this experimental setting of urban trees that are subject to very different contexts ?. I would suggest that the authors write in a more synthetic way.

**Authors response:** See answer above. Our experimental setup represents the 'ideal' case of an unobstructed solitary urban tree. This will result in a maximum water storage. A possibility to determine the influence of urban trees in more specific conditions (eg. A tall building 3 m on the south side of a tree in a wide east-west street lane) can be estimated through simulations using our findings as a maximum value. Moreover, because WetSpa takes LAI into account, a much broader variety of urban trees can be modeled. For example, a street tree will probably have a different LAI than a park tree.

**Change in manuscript:** See previous comment.

**Reviewer comment:** p 10 – paragraph 2.6: what difference between data processing and modelling (next section ?) It is not clear to me.

**Authors response:** We merged both sections, eliminated some explanations that are already known for a long time and wrote in a more synthetic way.

**Change in manuscript:** see manuscript section 2.6 'The Data Processing And Model Comparison'

**Reviewer comment:** p 12 – paragraph 2: If modelling and model comparison is a central part of the manuscript, I would suggest to present the models as part of the manuscript and not in Appendix.

**Authors response:** We decided to keep the equations in the appendix for multiple reasons: (1) the models are not new but have been developed a long time ago, (2) we don't change the functionality (equations) of the models , and (3) the readability of the manuscript. We add the equations to the appendix for the reproducibility of our results but believe that the equations have no other added

value to the manuscript as the differences between the models is discussed in the method section (paragraph 2.6).

**Change in manuscript:** No changes made.

**Reviewer comment:** p 12 – paragraph 2.8: Do these models apply to individual trees, as the field experiment, or a tree plant cover? I suppose that there is a very important difference between the two. It is a significant point concerning the model relevance.

**Authors response:** Gash and Rutter developed their models on forest stands whereas WetSpa is not specialized but was built to work at a regional basin scale (therefore the simplifications). However, we adapt the interception module in WetSpa to include LAI and set-up a model representing the V-catchment experimental set-up. We agree that it is important to note that Gash and Rutter were developed on a forest stand and also discuss the possible effects on the outcome in the discussion section.

**Change in manuscript:** p 12- paragraph 2.6: we added the text 'The Gash and Rutter model have been developed for a forest stand whereas the WetSpa model is adapted for a solitary tree' to clarify the differences which are also summarized in table 4.

**Reviewer comment:** p 13: the scenario analysis appears as important as the modelling and model comparison. Is it really the case, or is it just an illustration example, in which case, I would suggest to reduce its length.

**Authors response:** The scenario analysis indeed is used as an illustration example to emphasize the potential of urban trees in reducing runoff. To reduce the length of the manuscript we decided to remove the scenario analysis from the paper and instead discuss the potential impact of urban trees in the hydrological cycle in the discussion section (section 4.3 'The Potential Benefit Of Trees In An Urban Context').

**Change in manuscript:** see section 4.3

**Reviewer comment:** p 15 – lines 5 to 14: I would suggest to move this lines to Section 2.

**Authors response:** We agree and moved section 3.1  to Material and Methods (section 2.2).

**Change in manuscript**: See section 2.2.

**Reviewer comment** : p 17 – figure 8: I am not convinced that this figure is very useful here, it could eventually be moved in Appendix.

**Authors response:** We agree that this figure takes up to much space. We removed the figure (it is available upon request) and we now describe the rain event characteristics in section 2.4.

**Change in manuscript:** See section 2.4.

**Reviewer comment:** p 19 – paragraph 3.3: is a full page needed to see the weak sensitivity? A few lines would be enough.

**Authors response:** We agree, we removed the figures and shortened this section substantially. This part is now incorporated in section 2.5.

**Change in manuscript:** See section 2.5.

**Reviewer comment:** p 20: line 1-4: small events of 1 mm are not very important from a hydrological point of view

**Authors response:** We agree, this fact together with the high uncertainty associated with events < 1 mm led us to the decision to remove them from further analyses.

**Change in manuscript:** See section 2.4.

**Reviewer comment**: p 20: figure 10: the presence of the inflexion point on the regression lines is not obvious. Is it justified by physical reasons?

**Authors response:** A disadvantage of the Leyton method is that the inflection point is determined subjectively. We looked at the residual values of the $P_g$ vs $T_f$ plot to determine this point (not shown in manuscript), a sudden change in residual values indicated the inflection point.

We decided however to remove the Leyton analysis for interception storage capacity determination from the manuscript for several reasons:

1) The above described inherent subjectivity of the method.
2) The Letyon method cannot take into account seasonal changes of leaf cover. As our measurements periods encompass transition periods from leaf off to leaf on , this confuses the Leyton analysis.
3) A few large events can significantly alter the analysis.

For these reasons we decided to abandon the Leyton method as a means of comparison with our chosen method to determine interception storage capacity (Equation 5, Gomez et al., 2001). The Leyton method is still used for the free throughfall coefficient calculation, as shown in appendix A.

**Change in manuscript:** See manuscript section 2.2.

**Reviewer comment:** p 22: Equation 6. The authors introduce I as interception (in mm). The legend of Table 5 mentions the Interception storage capacity (S in mm), as Equation 5 (p 11) and in Appendix A (Eq. 1) I is the interception storage, and S the crown surface storage capacity, and IS appears in following equations. In equation 6, D is duration and in A-1, it is drip-off ..... A very careful checking of notations is required. What is the difference between interception, interception storage and storage capacity.

**Authors response:** We carefully checked our notations in the manuscript and made some changes. The definitions of interception, interception storage and interception storage capacity are explained in section 1.3.

In short: interception is a process, interception storage (I) is the total volume of water a tree can hold during an event that does not reach the ground surface and the interception storage capacity (S) is the maximum volume of water a tree can hold for a given time. The interception storage can be greater than the interception storage capacity when there are dry periods during a rainfall event and water can drip off or evaporate.

**Change in manuscript:** Definitions in section 1.3

**Reviewer comment:** p 22: Equation 6 – I am sceptical concerning the regression model (eq. 6) which provides better results, but which introduces 9 numbers, that is 9 parameters, which means that this model is strictly limited to the analysed data set, without any generalization. Such a model is of a very limited interest.

**Authors response:** We decided to remove the regression analysis from the manuscript. One reason for this is the comment mentioned by the reviewer. The other reason is that it allows us to focus more on the comparison between the standard forest canopy interception models (Gash- and Rutter) and the WetSpa, solitary tree model. We think this makes our manuscript more clear.

**Change in manuscript:** See manuscript

**Reviewer comment:** p 23: Figure 12 – Interception storage – A new denomination?

**Authors response:** This is the interception storage as defined in section 1.3, the total volume of water intercepted by the tree for an entire rainfall event.

**Change in manuscript:** See section 1.3

**Reviewer comment:** p 25: I would suggest that Figure 13 and Table 6 to be the central points of the model comparison.

**Authors response:** We agree and structured the text more around this table and figure. These are now Table 7 and Figure 6.

**Change in manuscript:** See manuscript

**Reviewer comment:** p 26-27 paragraph 3.7: I wonder about the usefulness of this paragraph "land cover change scenario" in this manuscript. In the light of previous results, it is not surprising that an increase of sealed surfaces generates a runoff increase, and that tree planting generates an increase of water storage.

**Authors response:** To make the focus of our manuscript clearer and to limit its length, we decided to remove the scenario analyses. We did add a section discussing the relevance of our findings to the urban context (section 4.3).

**Change in manuscript:** See section 4.3

**Reviewer comment:** p 27: It seems to me that interesting discussion subject could have been the following: i) validity of this model assessment given the possible differences between individual trees and tree plant cover; ii) application of the models to a more usual urban context: available data, types of trees, ...

**Authors response:** We restructured the discussion in a way that it better reflects and emphasizes our research questions. It is now structured as follows:

1) Comparison of our gathered dataset with datasets from other studies.
2) Comparison of model performance between the standard forest interception models and the individual tree WetSpa model.
3) Relevance of our experimental setup and model results to an heterogeneous urban environment.

**Change in manuscript:** See discussion section

**Reviewer comment:** p 32: Table to remove, it is unreadable (the data set can be provided on request).

**Authors response:** We removed the table for the manuscript.

**Response Reviewer 2**

**Authors:** We would like to thank reviewer 2 for his time to review our paper, we think his constructive comments have significantly improved the quality of our manuscript.

**Reviewer main comment 1:** The manuscript is, however, unusually long, and contains extended reviews of basic and standard canopy interception concepts that are not necessary and make the original contributions of the work difficult to extract. For example, Figures 1 and 5 are unnecessary: these have been standard for 50 years, and also the same analysis as Figure 5 was done for this research and presented as Figure 10. Sections 1.2, 1.3, and 2.6 could all be reduced substantially.

**Authors response:** We restructured the paper and shortened the manuscript. The main focus is now on the original contributions of the work: 1) The experimental setup and gathered dataset 2) Comparison of the performance of on one hand, the standard specialized forest interception models (Gash and Rutter) and on the other hand an adapted water balance model (WetSpa).

**Change in manuscript:** The regression analysis was removed because of its narrow application. The scenario analysis was also removed and instead a section (4.3) was added to discuss the relevance of our research to an urban context. Several paragraphs and subsections were removed or merged (e.g. section 3.3 'The sensitivity analysis' was substantially reduced and incorporated in sections 2.3 'The V-catchment Design' and 2.5 'The Meteorological Stations). Figures 1, 5, 6, 8, 9, 10, 11, 14 and 15 were removed.

**Reviewer main comment 2:** The choice of and behavior of WetSpa for interception modeling is difficult to evaluate. The WetSpa equations in Appendix A contain symbols that are not defined and the cited source for these is a white paper not contained in the References. As far as I can determine the simple assumptions it makes have never been tested in the refereed literature, despite the citations to support use of the WetSpa interception formulation (e.g., P5L15) that do establish that it has been used. There appears to be no scientific reason to include this model in a comparison, and it seems likely that it was chosen because of its familiarity to the authors.

**Authors response:** We want to test the performance of a water balance model to simulate interception storage of a solitary tree in order to be able to use the model in an urban context. WetSpa has been chosen due to the flexibility of the model which makes it easy to adapt it to our purposes (including LAI and setting up a V-catchment). The model flexibility and V-catchment set-up is well described in the PhD thesis of Elga Salvadore (Salvadore, 2015). The inclusion of LAI in the interception calculation is described in Wirion et al., 2016.

**Change in manuscript:** To better describe why we include WetSpa in the analysis we changed Paragraph 1.5 'Interception Models' and paragraph 1.6 'Research Questions'.

**Reviewer main comment 3:** The simplicity of the WetSpa formulation as an ultra-simple bucket model with, e.g., no provision for drip when storage is less than capacity or reduced evaporation rates from partially wetted canopy, is an interesting test against the more sophisticated Gash and Rutter models. That it appears to give empirically similar or superior results is a very useful finding for canopy interception science. However, the discussion makes essentially no attempt to explore the ramifications of this finding, either from a utilitarian (e.g., "should we be using simpler models") or physical (e.g., "what does it mean that only the coarsest components of the water budget need modeling?" Or "Is the superior performance of the super-simple model a fluke of this environment or should we be applying it more generally?") perspective. Re-stating and discussing the "WetSpa" formulation in terms of other, similar models in the older literature would help it to make the biggest contribution to the canopy interception literature.

**Authors response:** We agree that the good performance of WetSpa is surprising and important to notice. Further, we believe that different models serve different purposes. The disadvantage of Gash and Rutter in this case is that they were developed on forest stands and we believe this might affect their performance for bigger rainfall event as the evaporative potential differs in an urban context. We elaborate on this in the discussion of the results. We still promote the use of more specialized interception models for a more detailed understanding of interception on solitary trees. In the context of urban management, however, WetSpa seems a good alternative to quantify the interceptive potential of urban trees as a starting point to further analysis of other water balance components such as infiltration and runoff. When it comes to hydrological modelling the potential of interception storage is usually underestimated, simplified or even disregarded as its potential for flood mitigation is low. However, our study shows that most rainfall events in our climate are moderate and that 38% of the rainfall water during rain events is intercepted. It is thus important to consider interception in hydrological simulations and therefore we propose a simpler approach such as the one WetSpa uses.

**Change in manuscript:** Paragraph 4.2: Discussion on the model comparison.

**Reviewer main comment 4:** I think the urban hydrology modeling is a separate topic that is best left for another paper. Removing it would help shorten the paper to a more manageable size and also allow the strengths of the interception data to be better emphasized. Perhaps it is only a matter of taste, but I think the style in which the urban hydrological modeling text is written suggests bias on the part of the authors about the importance of urban trees, when a dispassionate evaluation would be more effective

**Authors response:** We agree with the reviewer's comments and removed the scenario analysis from the manuscript to focus more on the gathered dataset and the model comparisons.

**Change in manuscript:** Removal of the scenario analysis and addition of section 4.3 'The Potential Benefit Of Trees In An Urban Context' in the discussion, where the relevance of our findings in an urban context are discussed.

**Reviewer main comment 5:** The canopy interception conclusions mostly consist either of simple data or bland inferences based more on previous understanding of urban hydrology than on the results of this work. Adding theoretical discussion and making theoretical inferences from the data would help readers understand the scope of the work beyond the immediate context of this experiment.

**Authors response:** We have rewritten the conclusions with the aim to better emphasize our contributions to the urban hydrology literature. Furthermore, we elaborated on the model comparisons in the discussion section.

**Change in manuscript:** Our main conclusions are:

1) Both trees intercepted around 38% of gross precipitation, emphasizing the importance of (1) interception storage for reducing net rainfall and (2) accounting for interception storage in an urban water balance model.

2) The water balance model (WetSpa) and the specialized interception models of Gash and Rutter showed a similar performance when compared to the measurements. The three models underestimate interception storage for bigger rainfall events which we relate to a poor understanding of the evaporative behavior of intercepted rainwater during rain events in an urban environment. However, the relatively good performance of WetSpa for bigger rainfall events, its simplicity and its water balance framework promote it as a tool for assessing the interceptive potential of urban trees.

**Line comments**

**Reviewer comment:** P12L15 Salvadore et al. 2015 published a general review of models of urban hydrology, and it is not clear how that review supports this work.

**Authors response:** We agree, her PhD is a better reference on why we use the WetSpa model.

**Change in manuscript:** The sentence has been deleted as the information is redundant (cf: introduction).

**Reviewer comment:** Figure 2 is not needed; its function is duplicated by Figure 3.

**Authors response:** We decided to keep Figure 2 as it gives a good overview of the environment and near surroundings of the trees. Figure 3-4 are presenting a clear image of the schematic- and actual experimental construction.

**Change in manuscript:** None

**Reviewer comment:** Section 3.1 should be Methods

**Authors response:** We agree and moved this section to Methods. It is incorporated in section 2.2 and 2.4.

**Change in manuscript:** See section 2.2 and 2.4

**Reviewer comment:** Sections 2.5.2 and 3.3 the sensitivity analysis is difficult to understand. It seems like the data in the two panels of Figure 9 probably came from the same assumptions, but I cannot follow the text P9L34-37

**Authors response:** We removed Figure 9 because it took up too much space and wrote the sensivity analysis more concisely. This part was added to section 2.3 and 2.5..

**Change in manuscript:** See section 2.3 and 2.5.

**Reviewer comment:** Figure 10 the inflection point is plotted at ~2.5 but listed as 4.72.

**Authors response:** Because we decided to remove the Leyton analysis to estimate interception storage capacity from the manuscript, Figure 10 was also removed. A modified version of this figure, used to determine the free throughfall coefficient, can be consulted in Appendix A.

**Change in manuscript:** See manuscript

**Reviewer comment:** Figure 13e is unnecessary. The same information is in a-d. The equations for the regressions should be presented

**Authors response:** We added the regression equations and $R^2$ on the plot area and removed Figure 13d (the regression) and Figure 13e from the manuscript.

**Change in manuscript:** See figure 13 in the manuscript

**Reviewer comment:** Table 2 I disagree that regression does not account for evaporation during the event. It simply does so implicitly.

**Authors response:** We agree with this comment but decided to remove the regression from the manuscript because of its limited applicability.

**Change in manuscript:** See manuscript

[revised manuscript text omitted]

$\text{VPA}_{constr}$, $\text{VPA}_{free}$, $\text{VPA}_{cont}$, $\text{VPA}_{st}$ are the vertical projection areas of the whole V-catchment construction, the part of the construction not covered by the tree, the catchment container and the stemflow container respectively. $\Delta H_{cont}$ and $\Delta H_{St}$ are the height differences recorded after a rain event in the catchment- and in the stemflow container.

Not taking the tree interception into account, the catchment container reached its maximum capacity (493 L) when a rain event of 7.5 mm occurred. To take into account larger rainfall events, we followed the procedure explained below:

Each rain event that filled the catchment container was divided in two parts: the first part lasts until the container is filled and the second part starts when the container is full and all additional rain overflows to the ground. If the amount of rain fallen until the moment the container filled was larger than the interception storage capacity of the tree, it was assumed that the interception storage capacity was reached and all additional water would be converted to throughfall. Otherwise the event was discarded. The throughfall values of the first and second part of the event are then summed and compared to the interception storage values of the first part of the event. The stemflow container never filled completely and was analyzed on a whole event basis.

The interception storage capacity (S; mm) was calculated with an empirical equation based on LAI (-) (Gómez et al., 2001):

**(5)**

$$S = 1.184 + 0.490\,LAI - R^2 = 0.76$$

~~For comparison, we also used the method developed by Leyton et al. (1967) to calculate the interception storage capacity. A line of unit slope, minus the stemflow percentage, is drawn on a gross precipitation vs throughfall plot through those rain events where evaporation is assumed minimal. Only events where $P_g$ is large enough to fill the storage capacity are used. The determination of the amount of precipitation to reach canopy saturation is subjective and based on the recognition of an inflection point on the graph. To make recognition of this inflection point easier, the residuals of the $P_g$ vs $T_f$ regression were plotted against the $P_g$. The inflection point is recognized by a sudden change in the variability of residual values. The interception storage capacity is then found by the negative intercept of this upper envelope line with the y-axis. The upper envelope represents events with minimal evaporation. The points left of the inflection point represent events where rainfall was insufficient to fill the canopy storage completely. The parameter of the gross precipitation of a regression through the lower envelope of these points represents the free throughfall coefficient. An example of this method from (Sadeghi et al., 2015) is shown in Fig. 5:.his methodabove(,because equation 5 takes into account seasonal changes by using LAI as independent variable. Both our measurement periods (Table 1) span transition periods between leaf-on and leaf-off, which makes the use of the Leyton method to calculate interception storage capacity less preferable. Other downsides of the Leyton method are its inherent subjectivity and possible skewing of results through one or two very large rain events.~~

[revised manuscript text omitted]
 an integrated water balance framework, the influence of individual city trees on the whole urban balance can be accurately simulated.The Gash model performs better than the Rutter model on the individual tree level in temperate climates. The performance of WetSpa is in between both interception models and best for bigger rainfall events ($P_g$ > 10 mm).~~

(3) The regression equation we developed for the interception storage in temperate climates was found to be tree- and season- specific. We therefore don't recommend its use for a wider application as its performance is limited by the amount of meteorological data and by the tree architectural characteristics.

(4) The scenario analysis shows that even small scale construction sites can have an impact on the water management in cities. In our scenario a green space is replaced by new student houses on the VUB campus in Brussels, Belgium. This change increases net rainfall by 20 % and surface runoff by 49 %. A more sustainable scenario where all areas around buildings are covered by trees re-establishes the interception storage and thus limits net rainfall. However, due to less pervious surfaces, infiltration is still limited and surface runoff increased. This scenario analysis gives an indication of how city trees retain rainfall water and limit surface runoff production.

2)

**Appendix A: Free throughfall coefficient estimation:**

Leyton graphs for the Norway Maple (all year n= 39) and small leaved lime (all year n=25):

[Figure]

$$\text{Tf (mm)} = -0.63 + \mathbf{0.54} \text{ X Pg (mm)} \qquad \text{Tf (mm)} = -0.57 + \mathbf{0.30} \text{ X Pg (mm)}$$

| Norway Maple | Free throughfall coefficient (p) |
|---|---|
| All year (n=39) | 0.54 |
| 1st measurement period (n=13) | 0.6 |
| 2nd measurement period (n=26) | 0.32 |
| Small leaved lime | |
| All year (n=25) | 0.3 |
| 1st measurement period (n=8) | 0.47 |
| 2nd measurement period (n=17) | 0.25 |

**Appendix B: Model equations:**

$$I = S + E = P_g - (T_f + D + S_f)$$

equation 1 (Xiao, 1998)

$I$: interception storage [mm]
$S$ : crown surface storage capacity = interception storage capacity [mm]
$E$ : evaporation [mm]
$P_g$ : gross rainfall [mm]
$T_f$: free trough-fall [mm]
$D$ : drip-off [mm]
$S_f$: stemflow [mm]

$$T_f = P \times P_g \qquad\qquad\qquad \text{equation 2 (Xiao, 1998)}$$

$P$ = gap fraction [-]

[Figure]

$$D = 0 \qquad \text{for } I < S \qquad \text{equation 3 (Valente et al.,}$$
1995)
$$D = I - S \qquad \text{for } I \geq S$$

5 **Gash**

$$I = (1\text{-}p) \times P_g \qquad \text{for } P < P' \qquad \text{equation 5 (Gash, 1979)}$$
$$I = (1\text{-}p) \times P' + \frac{Ea}{R} \times (P_g\text{-}P') \qquad \text{for } P > P'$$

$Ea$ : mean evaporation / canopy cover [-]
10 **R**: mean rainfall/ saturated canopy cover [-]
**P'**: precipitation reaching canopy saturation [mm]

$$P' = -\frac{R}{Ea} \times S \times ln(1 - \frac{Ea}{R \times (1-p)}) \qquad \text{equation 6}$$
(Gash, 1979)
15
**Rutter**

$$P_{soil} = T_f + D + S_f$$

20 and

$$P_{soil} = P_g \text{ - E- (S-IS)} \qquad \text{for P - E > C – IS}$$
$$P_{soil} = 0 \qquad \text{for P - E <= C - IS} \qquad \text{equation 7 (Vegas et al., 2012)}$$

25 **P$_{soil}$ :** precipitation reaching the ground (net precipitation) [mm]
**IS:** interception storage at timestep before actual rainfall [mm]

$$E = \frac{IS}{S} \times PET \qquad \text{equation 8 (Vegas et al., 2012)}$$

30 **PET**: Potential evapotranspiration estimated with the Penman-Monteith equation (Monteith, 1965).

$$I = E + (S\text{- IS}) \qquad \text{equation 9 (Vegas et al., 2012)}$$

**WetSpa**

35 $$I = S – IS \qquad \text{for P > C – IS} \qquad \text{equation 10 (WetSpa manual,}$$
2009Liu et al., 2004)
$$I = P_g \qquad \text{for P <= C- IS}$$

and
40 $$IS (t) = IS (t\text{-}1) + I (t) – E (t) \qquad \text{equation 11 (Liu et al.,}$$
2004WetSpa manual, 2009)
and

$$E(t) = IS(t\text{-}1) \qquad \text{for PET > IS(t-1)} \qquad \text{equation 12 (Liu et al.,}$$
2004WetSpa manual, 2009)

$$E\ (t) = PET \qquad \text{for PET} < IS(t-1)$$

**Appendix B: equations Figure 9.**

[revised manuscript text omitted]

---

## Author Response (AR2)

Authors Response: report of Referee #2

To the editor:

Thank you for giving us the chance to explain our reasoning for adding the large rain events.

As suggested by the editor, we added our reasoning in the manuscript in both the M&M section (P12L11-16) and also in the result section (P13L8-P14L5). An extra appendix (B) is added to give the reader more insight in the events in question.

Full comment Referee:

In my original review, I was lost in the excess information and missed a crucial problem: all net precipitation data have been assumed for all events larger than 7.5 mm because of instrument failure (P10L10). Because of this, there are not enough actual data to support empirical estimates of canopy storage and all data-based estimation of canopy storage has been abandoned in this revision in favor of a purely empirical equation derived off site. These problems defeat most of the purpose of the experiment. By assuming storage capacity and also assuming that total storm evaporation loss is equal to that storage capacity, the experiment is left with insufficient data on canopy interception by which to calibrate models. Canopy storage capacity is one of the most important variables in the Rutter-Gash formulation, so those models cannot be used. The consequences for the assumptions are also obvious in the data such as figure 5, where there is a clear inflection point near 7.5 mm. The assumptions thus clearly drive the results, and the conclusions cannot be considered supported by data.

We would like to thank the reviewer for critically reviewing our manuscript. We understand his concern but disagree on the idea that our method is assumption driven, as we back up the assumptions with our measurements. We try to respond to the reviewer's reasoning below. Please let us know if any doubts about our approach remain and what would be needed to improve our manuscript.

"all net precipitation data have been assumed for all events larger than 7.5 mm because of instrument failure (P12L10). "

It is correct that when cumulative precipitation exceeded 7.5 mm, we assumed that all subsequent rain would be converted to through fall or stem flow (net precipitation). This is not due to instrument failure but due to a limitation in the size of our rainfall collection system. The reviewer is correct in his concerns that if intra-event evaporation occurred (during intermitted short periods of drought in a rain event) after this 7.5 mm threshold, this intra-event evaporation would not be detected with our measurement system, which would give rise to an underestimation of canopy storage and an overestimation of through fall. However all these large events were thoroughly checked on the possibility of intra event evaporation. The amount of precipitation significantly exceeded 7.5 mm in 10 out of 64 rain events. In Appendix B of the revised MS, the time-intensity and cumulative graphs of these 10 events are depicted to detect intermittent dry periods (ranked largest – smallest event). For some events with dry periods occurring late in the rain event, we also added humidity and temperature graphs to provide extra information. These intermitted dry periods would be problematic if they occurred during sunlight hours and late in the rain event when the threshold of 7.5 mm is exceeded. This is the case in 2 out of 10 events (event 57 and event 33). However only 0.33 mm and 0.23 mm of rain fell after the intermittent drought period of these events. This amount is, in our opinion, negligible in comparison with the total amount of PP that has fallen during these events (17,78 mm and 10.81 mm which makes the rain fallen after the drought period 1.8% and 2,1% of total PP respectively).

Because of this, we think our assumption for larger events is correct and that they can be used in the analysis.

"Because of this, there are not enough actual data to support empirical estimates of canopy storage and all data-based estimation of canopy storage has been abandoned in this revision in favor of a purely empirical equation derived off site. "

It is true that we abandoned the Leyton method (data-based estimation) for defining the canopy storage. After the first review, where we received comments from both reviewers about the definition of the inflection point, we decided to remove the Leyton analysis from the manuscript. The main reasons are the subjectivity of the method (Klaassen, 1998; Link et al.,2004) and the fact that seasonal changes are not taken into account. Therefore we decide to use the empirical equation of Gomez. It is true that the Gomez equation has been derived off-site but it is based on the leaf area index which we measured for all trees and seasons. Several authors (De Jong & Jetten, 2007; Galdos et al., 2012) point out the importance of using leaf area index to estimate interception storage capacity. We therefore don't agree that we assume interception storage capacity as it is based on an equation using measured leaf area index values from our trees of interest. The Gomez equation has been used for broadleaf trees in other publications (Verbeiren et al., 2016 & Wirion et al., 2017).

"These problems defeat most of the purpose of the experiment. By assuming storage capacity and also assuming that total storm evaporation loss is equal to that storage capacity, the experiment is left with insufficient data on canopy interception by which to calibrate models. "

As we stated above, only two out of 10 events larger than 7.5 mm had intra event dry periods during sunlight hours after the 7.5 mm threshold was reached. Because very little rain fell after these drought periods, the error margin of our assumption is very low.

"Canopy storage capacity is one of the most important variables in the Rutter-Gash formulation, so those models cannot be used. The consequences for the assumptions are also obvious in the data such as figure 5, where there is a clear inflection point near 7.5 mm. The assumptions thus clearly drive the results, and the conclusions cannot be considered supported by data…."

We are not sure to which figure 5 the reviewer refers to. If it is figure 5 (Figure 5: Example of the Leyton method from (Sadgehi, 2015).) of the first version of our manuscript, it is a literature based description of the Leyton method and is not at all related to our data. I agree with the reviewer that we can see an inflection point at the 7.5mm threshold however there is no relation at all to our dataset. If the reviewer refers to the figure 5 ( Figure 5: Interception storage vs gross precipitation for all events for both trees.) of the revised manuscript, we don't understand that comment as there is no inflection point on that figure. This figure shows a larger variation in interception storage with larger events.

Thank you very much for considering our explanations.

Best regards,

Vincent and Charlotte

References:

De Jong, S. M., & Jetten, V. G. (2007). Estimating spatial patterns of rainfall interception from remotely sensed vegetation indices and spectral mixture analysis. International Journal of Geographical Information Science. doi:10.1080/13658810601064884

Vegas Galdos, F., Álvarez, C., García, A., & Revilla, J. a. (2012). Estimated distributed rainfall interception using a simple conceptual model and Moderate Resolution Imaging Spectroradiometer (MODIS). Journal of Hydrology, 468-469, 213–228. doi:10.1016/j.jhydrol.2012.08.043

Klaassen, W., Bosveld, F. and de Water, E.: Water storage and evaporation as constituents of rainfall interception, J. Hydrol., 212–213, 36–50, doi:10.1016/S0022-1694(98)00200-5, 1998.

Link, T. E., Unsworth, M. and Marks, D.: The dynamics of rainfall interception by a seasonal temperate rainforest, Agric. For. Meteorol., 124(3–4), 171–191, doi:10.1016/j.agrformet.2004.01.010, 2004.

Verbeiren, B., Khanh Nguyen, H., Wirion, C., & Batelaan, O. (2016). An Earth observation based method to assess the influence of seasonal dynamics of canopy interception storage on the urban water balance. Belgeo, (2), 0–21. doi:10.4000/belgeo.17806

Wirion, C., Bauwens, W., & Verbeiren, B. (2017). Location- and time-specific hydrological simulations with multi-resolution remote sensing data in urban areas. Remote Sensing, 9(7). doi:10.3390/rs9070645

---

## Author Response (AR3)

**Comment from editor:**

Dear authors,

I consider the paper close to be published in ESS. However, you need to adjust the figure to be in line with the typical scientific style in HESS. So, please (1) remove all border lines of your figures, Increase font size of the labels, use darker grey or black for labels and lines, align figures, optimise the legend (do not repeat), make time series readable (in particular in the supplemental materials). In general, Excel is not a good approach to generate figure to scientific publication without further optimisation.
The conclusion reads more like an abstract - please adapt accordingly.
Change numbering and symbols of equations. For example a symbol called PET or IS is not accepted in HESS.
Some minor points: table 7 is difficult to read - rearrange. The reference to some tables is not correct (Table 7 7) page 15.
In general please read the information for authors in detail and change your manuscript accordingly, either wiese I cannot accept it for publication in HESS.

Best regards
Markus

**Author's response:**

We made the following changes according to the manuscript preparation guidelines for authors:

- We made improvements to the figures 2, 5, 6 and B1 in accordance with your remarks and with the Authors' guidelines. The vector figures are in .png format in the text but we have them in .eps and .pdf format as well. We will upload these when requested.
- Labeled and cited figures, equations, symbols, ect.. according to the Authors' guidelines.
- Tables are moved to a separate sheet after the references according to the Authors' guidelines.
- Table 7 is made easier to read.
- Conclusion is rewritten

If you have any further remarks please let us know.

Kind regards,

Vincent and Charlotte